# Lifelong Test-Time Adaptation via Online Learning in Tracked Low-Dimensional Subspace

**Dexin Duan, Rui Xu, Peilin Liu, and Fei Wen**[*]

School of Integrated Circuits, School of Information Science and Electrical Engineering
Shanghai Jiao Tong University, Shanghai, China, 200240
{jumpywizard,Rui_Xu,liupeilin,wenfei}@sjtu.edu.cn

## Abstract

Test-time adaptation (TTA) aims to adapt a source model to a target domain using only test data. Existing methods predominantly rely on unsupervised entropy minimization or its variants, which suffer from degeneration, leading to trivial solutions with low-entropy but inaccurate predictions. In this work, we identify *entropy-deceptive* (ED) samples, instances where the model makes highly confident yet incorrect predictions, as the underlying cause of degeneration. Further, we reveal that the gradients of entropy minimization in TTA have an intrinsic low-dimensional structure, driven primarily by *entropy-truthful* (ET) samples whose gradients are highly correlated. In contrast, ED samples have scattered, less correlated gradients. Leveraging this observation, we show that the detrimental impact of ED samples can be suppressed by constraining model updates within the principal subspace of backward gradients. Building on this insight, we propose LCoTTA, a lifelong continual TTA method that tracks the principal subspace of gradients online and utilizes their projections onto this subspace for adaptation. Further, we provide theoretical analysis to show that the proposed subspace-based method can enhance the robustness against detrimental ED samples. Extensive experiments demonstrate that LCoTTA effectively overcomes degeneration and significantly outperforms existing methods in long-term continual adaptation scenarios. Code is available at https://github.com/ThunderDavid/LCoTTA.

## 1 Introduction

Test-time adaptation (TTA) aims to address domain shifts by adapting the model to the target domain during inference, without requiring access to source data [1, 2, 3, 4, 5, 6, 7, 8]. This need for on-the-fly adaptation arises broadly across other tasks under distribution shifts[9, 10, 11, 12, 13], motivating TTA as a general deployment-time solution. Unlike traditional domain adaptation methods, TTA operates in a fully unsupervised manner. It leverages only the unlabeled test data received during inference, enabling the model to adaptively improve its performance when faced with domain shift on-the-fly. As a fully unsupervised online learning paradigm, TTA methods predominantly rely on unsupervised learning objectives, such as entropy minimization or its variants [2, 14]. However, such methods are prone to instability. For instance, entropy minimization suffers from a trivial solution that assigns all predicted probability to the most probable (but may be incorrect) class. Consequently, these methods are susceptible to performance degeneration during continual adaptation.

To address this, various methods have been proposed. For example, [2] leverages entropy minimization at the batch level while restricting updates to only the normalization parameters. Moreover, some approaches employs teacher-student networks, where a teacher model provides stable guidance through a moving average of parameters to prevent collapse [8, 6, 4]. Furthermore, model

---

[*]Corresponding author: Fei Wen, email:wenfei@sjtu.edu.cn

39th Conference on Neural Information Processing Systems (NeurIPS 2025).

resetting strategies, which periodically reset partial or full model parameters to their initial states, have demonstrated effectiveness in mitigating error accumulation during adaptation [8, 7, 15, 16]. Generally, these strategies have shown effective under the current mainstream evaluation protocol, which typically involves single-epoch adaptation on a test set. However, in more realistic lifelong scenarios, where the model is required to adapt continually over the long term, these methods still face challenges in maintaining consistently robust performance.

Although numerous heuristic approaches have been proposed to mitigate the degeneration, an in-depth analysis of its underlying cause remains lacking. In this work, we delve into the underlying cause of the degeneration problem and provide novel insights to address it at its core. Specifically, we show that a main underlying cause of the degeneration in entropy-minimization based methods is the presence of *entropy-deceptive* (ED) samples, where the model produces highly confident yet incorrect predictions. These samples frequently arise when facing out-of-distribution data in the TTA setting, which mislead model updates and exacerbate error accumulation over time, especially in long-term continual adaptation scenarios.

Furthermore, we show that the backward gradients of the entropy loss in TTA exhibit an intrinsic low-dimensional structure. It is primarily driven by *entropy-truthful* (ET) samples, whose gradients are highly correlated as they tend to share similar update directions. In contrast, gradients from ED samples are scattered and less correlated. Based on this observation, we present a nontrivial finding that, the detrimental impact of ED samples can be suppressed by constraining weight updates in a low-dimensional principal subspace of the gradients. Building on this insight, we propose LCoTTA, a lifelong continual TTA method that tracks the principal subspace of gradients online and utilizes their projections for adaptation. This approach can effectively suppress the impact of ED samples and ensures robust performance in challenging long-term continual adaptation scenarios. In summary, the main contributions are as follows:

- An observation that the degeneration of the entropy minimization based TTA method in continual adaptation is primarily caused by ED samples, instances where the model makes highly confident but incorrect predictions.

- A novel finding that, without using any supervision information, the detrimental impact of ED samples can be suppressed by exploiting the intrinsic low-dimension structure of the gradients, which is primarily formed by the correlated gradients of ET samples.

- A lifelong continual TTA method LCoTTA, which tracks the principal subspace of gradients online and utilizes their projections into this subspace for adaptation. LCoTTA does not suffer from degeneration and enables robust long-term continual adaptation.

- A theoretical analysis that shows the proposed subspace projection based method can enhance the robustness against detrimental ED samples.

- Extensive experiments demonstrate that LCoTTA maintains robust performance in long-term continual adaptation scenarios and significantly outperforms existing continual TTA methods.

While this work focuses on TTA, the method of exploiting ET and ED gradient structure to effectively distinguish them in an unsupervised manner holds promise for broader applications in unsupervised learning tasks. The proposed method is complementary to existing approaches for robust TTA, such as teacher-student networks, model resetting, and weight regularization. Integrating it with these methods can be expected to further improve performance.

## 2 Related Work

**Test-time Adaptation.** [1] first proposed updating the activation statistics of BN to enhance model robustness. Further, MemBN [17] introduced a memory-based BN approach, which aggregates and adaptively weights stored statistics to achieve robust TTA. Tent [2], one of the pioneering works, updates only the affine transformation parameters of BatchNorm layers by entropy-minimization. SHOT [3] combines entropy minimization with diversity regularization to achieve robust TTA. DePT [18] leverages visual prompts to efficiently adapt to target domains and bootstrap source representations. Furthermore, single-sample methods have been explored in [19, 20], while non-parametric approaches are proposed in [21, 22]. Moreover, neuro-inspired method and energy-based model have been considered in [23] and [24].

**Continual Test-time Adaptation.** Continual TTA aims to adapt a model to dynamic and evolving target domains. CoTTA [8] introduces a teacher-student framework with consistency loss, while EcoTTA [6] employs meta-networks and self-distilled regularization to improve memory efficiency. TTACOPE [25] leverages supervised pretraining on labeled source datasets to improve initialization, while BECoTTA [26] utilizes a mixture-of-domain low-rank experts for domain-adaptive routing. Moreover, there exists a number of recent works designed for robust continual TTA [27, 28, 29, 30, 31, 32, 33, 34, 35, 36, 16, 37, 38, 39, 40]. Despite the effectiveness of model-resetting, teacher-student learning, and weight regularization approaches, they still face challenges in maintaining consistently robust performance in long-term continual adaptation scenarios. More importantly, there lacks an in-depth analysis on the underlying cause of degeneration in unsupervised TTA, which is crucial for understanding and mitigating it.

**Low-dimensional learning.** The low-dimensional structure of neural-network learning has been extensively studied [41, 42, 43, 44, 45, 46, 47]. These studies reveal that loss landscapes of neural-networks reside within an intrinsic dimension, which enables model weights to be optimized in a low-dimensional subspace. Unlike these works focusing on the low-dimensional structure of weights sampled along the optimization trajectory, we reveal the low-dimensional structure of batch-based stochastic gradients during the adaptation process, which is driven by ET samples in TTA and can help suppress the gradients of ED samples.

## 3  Analysis on the Degeneration of Entropy-Minimization Based TTA

### 3.1  Prelimineries

Let $f(x; \theta)$ be a model with weights $\theta$ pre-trained on a source domain $\mathcal{D}_s = \{(x_i, y_i)\}_{i=1}^N$, which follows a distribution $P_s(x)$. Given the source model $f(x; \theta)$, the goal of TTA is to adapt it to test data $\mathcal{D}_t = \{x_i\}_{i=1}^M$, where the data distribution $P_t(x)$ deviates from the training distribution, i.e. $P_t(x) \neq P_s(x)$. Without access to source data and without using any supervision information on test data, TTA methods typically utilize unsupervised losses, such as the entropy loss [2]

$$\mathcal{L}_e = -\frac{1}{B} \sum_{i=1}^{B} \sum_{c=1}^{C} \hat{p}_c(x_i) \log \hat{p}_c(x_i), \tag{1}$$

where $\hat{p}(x_i)$ is the model predicted probability on sample $x_i$. It encourages the model to make confident predictions by minimizing the entropy of the model predictions.

Traditional TTA methods typically focus on adaptation on test data over a limited number of steps, but real-world scenarios often involve continuously evolving domain shifts, requiring models to adapt continually over the long term. Lifelong TTA addresses this by adapting a model to sequentially arriving test data from evolving target domains $\{P_t : t = 1, 2, \cdots\}$. At each step $t$, the model predicts output $f(x_i; \theta_t)$ for input $x_i$ and updates its parameters for future steps, i.e., $\theta_t \to \theta_{t+1}$. This setting is challenging as models relying on unsupervised losses are prone to degradation over time.

### 3.2  Entropy Reliability of Test Samples

As discussed above, existing TTA methods, operating without access to source data and supervision on test data, predominantly rely on unsupervised entropy minimization or its variants [2, 8, 4]. While simple and effective, entropy minimization as an unsupervised loss can be unreliable and mislead model updates. Specifically, entropy minimization has a trivial solution that assigns all predicted probability to the most probable class. This is particularly problematic for samples with high-confidence but incorrect predictions, which frequently occur when the target domain significantly deviates from the source domain. For instance, given a ground-truth label [0,0,0,0,1], a model prediction of [0.1,0.2,0.4,0.1,0.2] optimized via entropy minimization is likely to converge to [0,0,1,0,0], resulting in incorrect predictions.

To address this degeneration problem, Tent [2] proposes to jointly optimize batched predictions while restricting updates to normalization parameters. However, this approach struggles to maintain satisfactory performance in continual adaptation. Although methods like model resetting and teacher-student frameworks [2, 8, 48, 49] partially mitigate degeneration, as shown in our experiments (Figure 4), they still suffer from performance degradation in long-term adaptation scenarios. Press

et al. [50] further show that EM first improves accuracy by embedding test data close to the class means of training data, but over many iterations, it pushes test embeddings far from the training data, resulting in degraded accuracy. Here, we offer a new perspective on EM by focusing on the *entropy reliability* of predicted test samples.

To better characterize the reliability of predictions, we introduce the Entropy Reliability Score (ERS). Let $\hat{\mathbf{p}} = [\hat{p}_1, \hat{p}_2, \ldots, \hat{p}_C]$ denote the model's predicted probability distribution over $C$ classes, where $\sum_{i=1}^{C} \hat{p}_i = 1$ and $\hat{p}_i \in [0,1]$. For a sample with ground-truth class $c \in \{1, 2, \ldots, C\}$, its ERS is defined as

$$S(\hat{\mathbf{p}}) := \hat{p}_c - \max_{i \neq c} \hat{p}_i, \tag{2}$$

where $\hat{p}_c$ is the predicted probability for the true class $c$, and $\max_{i \neq c} \hat{p}_i$ is the highest predicted probability among all incorrect classes. The ERS measures the confidence gap between the true class and the most competitive incorrect class, providing a quantitative indicator of prediction reliability. Based on ERS, we further define ET and ED samples.

**Definition 1:** Given a model prediction $\hat{\mathbf{p}}$ and with the definition of the reliability score in (2), a sample is classified as **entropy-truthful** if $S(\hat{\mathbf{p}}) > 0$; otherwise, it is classified as **entropy-deceptive**.

### 3.3 The Degeneration of Entropy Minimization

Here we further dive into the underlying cause of the degeneration problem in entropy-minimization based methods. We present an observation that this issue is primarily driven by ED samples. From Definition 1, ET samples indicate reliable predictions where the model assigns the highest confidence to the true class. In contrast, ED samples reflect unreliable predictions, where the model assigns higher confidence to incorrect classes, potentially misleading optimization processes such as entropy-minimization. This issue is particularly pronounced for highly confident ED samples, which exhibit low entropy despite being incorrect.

To analyze the impact of different types of samples on the adaptation process, we conduct experiments under a typical continual TTA scenario on the CIFAR100C and ImageNetC datasets with Gaussian corruption at severity-5. The samples are sorted by ERS, and pretrained ResNeXt-29 and ResNet-50 from [51] are adapted using entropy-minimization with different subsets of these samples. Among the samples ranked by ERS, the top **58.17%** on CIFAR100C and **11.59%** on Ima-

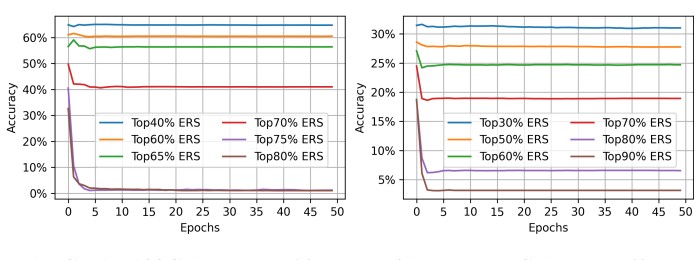

(a) CIFAR100C (ResNeXt-29)      (b) ImageNetC (ResNet-50)

Figure 1: Accuracy of entropy-minimization based continual TTA using different subsets of the samples. Among the samples ranked by ERS under Gaussian corruption, the top **58.17%** on CIFAR100C and **11.59%** on ImageNetC are identified as ET samples with $S(\hat{\mathbf{p}}) > 0$.

geNetC are identified as ET samples with $S(\hat{\mathbf{p}}) > 0$. Figure 1 shows the performance of continual adaptation over 50 epochs.

The results demonstrate the detrimental impact of ED samples on continual adaptation. For example, on CIFAR100C, when adaptation is performed exclusively on ET samples (e.g., the top 40% of samples), the model does not exhibit degeneration during continual adaptation. In contrast, expanding the sample set to include ED samples (e.g., the top 75% or 80% of samples) leads to significant degeneration. A similar trend is observed on ImageNetC.

## 4 Method

The above analysis suggests that the degeneration of entropy-minimization can be avoided by using only ET samples for adaptation. However, identifying ET samples without supervision is challenging. In this section, we reveal that the impact of ED samples can be suppressed by leveraging the low-dimensional structure of ET gradients. Then, we propose a robust lifelong continual TTA method.

## 4.1 Correlated ET Gradients Forming A Low-Dimensional Structure

As discussed above, we identified ED samples as the underlying cause of model degeneration. Here, we show that *gradients from ET samples are highly correlated, which form a low-dimensional subspace. In contrast, gradients from ED samples are scattered and less-correlated.*

Figure 2 (a) and (c) shows the correlation between the gradients of different batches on CIFAR100C and ImageNetC (Gaussian noise with severity-5) for ResNeXt-29 and ResNet-50, respectively. The samples are sorted by their ERS values from the most entropy-truthful to the most entropy-deceptive. It can be seen that gradients of ET samples are highly correlated, while gradients of ED samples are less correlated.

This phenomenon arises as gradients of ET samples tend to share similar update directions in parameter space, thus are correlated. In contrast, the gradients of ED samples are less-correlated, as their update directions are more scattered and do not have consistent directions. Based on this analysis, we can expect that gradients of entropy-minimization in TTA have a low-dimensional structure, which is primarily formed by the principal update directions shared by ET sample gradients.

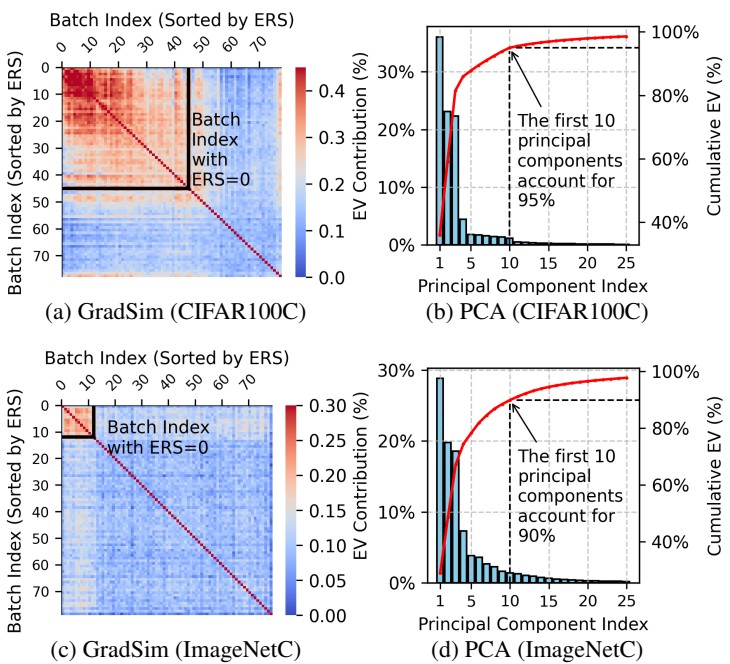

Figure 2: (a) and (c): Pairwise cosine similarity of batch gradients (GradSim) of test samples sorted by ERS. ET samples exhibit high gradient similarity, and the boundary (a sharp drop) in similarity closely aligns with the batch index where ERS=0. (b) and (d): Explained variance analysis of the gradient matrix $G$ during TTA.

Let $g_t = \nabla_\theta \mathcal{L}_e(\mathcal{B}_t; \theta)$ denote the gradient of the entropy loss $\mathcal{L}_e$ defined in (1) with respect to the model parameters $\theta$ on a batch of samples $\mathcal{B}_t$. To investigate the low-dimensional structure of stochastic gradients, we collect $T$ backward gradients $G = [g_1, g_2, \cdots, g_T] \in \mathbb{R}^{n \times T}$ over $T$ sequential batches sampled during the adaptation process, where $n$ is the number of model parameters. Principal component analysis (PCA) of $G$ is then performed using SVD.

Figure 2 shows typical PCA results in the TTA experiments on CIFAR100C (ResNeXt-29) and ImageNetC (ResNet-50). Gradients are sampled every 50 iterations with $T = 50$. The bar plot shows the explained variance of individual principal components, while the red curve represents the cumulative explained variance. Notably, the first few principal components dominate, with the top 10 components capturing more than 90% of the total variance. This result demonstrates that parameter gradients during TTA reside in a low-dimensional subspace, formed by ET sample gradients. In contrast, ED gradients do not have such a structure, as shown in Figure 9 in Appendix J.

## 4.2 Suppressing Entropy-Deceptive Samples via Low-Dimensional Gradient Structure

We reveal that ET gradients are predominantly concentrated within a low-dimensional principal subspace, whereas ED gradients exhibit a weaker alignment with this subspace. To show this, we first construct a $r$-dimensional principal subspace of the gradients. Specifically, given a collected

gradient matrix $G \in \mathbb{R}^{n \times T}$, we extract a $r$-dimension subspace of it via the following formulation

$$\max_{U_r \in \mathbb{R}^{n \times r}} \operatorname{tr}\left(U_r^\top G G^\top U_r\right), \quad \text{s.t.} \quad U_r^\top U_r = I, \tag{3}$$

where $U_r \in \mathbb{R}^{n \times r}$ contains the orthonormal bases corresponding to the largest $r$ eigenvalues of $GG^\top$, which spans the $r$-dimension principal subspace of $G$, with $r \ll n$. Formulation (3) is a standard PCA problem that can be solved by eigen-decomposition of $GG^\top$.

Given an extracted subspace spanned by $U_r$, we analyze the impact of gradient projection into this subspace. Specifically, for a gradient $g_t$, its low-dimensional representation is obtained via projection as $U_r^\top g_t$, and then back-projected to the original parameter space as

$$\tilde{g}_t = U_r U_r^\top g_t. \tag{4}$$

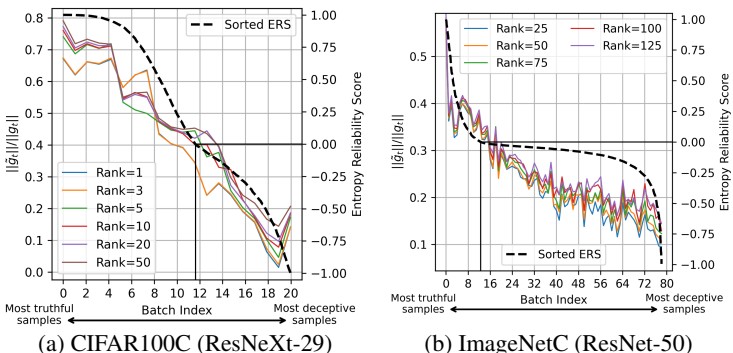

(a) CIFAR100C (ResNeXt-29)  (b) ImageNetC (ResNet-50)

Figure 3: The $\ell_2$-norm ratio $\frac{\|\tilde{g}_t\|}{\|g_t\|}$ for different sample types. The samples are sorted by their ERS values from most entropy-truthful (left) to most entropy-deceptive (right).

We then examine the effect of this principal subspace representation on ET and ED samples. Using the same experimental setting as in Figure 1 with the CIFAR100C and ImageNetC, we sort the test samples by their ERS values. With the sorted samples, we compute the $\ell_2$-norm ratio $\|\tilde{g}_t\|/\|g_t\|$, from the most truthful to the most deceptive samples. Figure 3 presents the results for different subspace dimensions, where the ratio $\|\tilde{g}_t\|/\|g_t\|$ is computed on batched samples.

Interestingly, as shown in Figure 3, the ratio $\|\tilde{g}_t\|/\|g_t\|$ generally decreases from the most truthful to the most deceptive samples. This implies that the principal subspace representation (4) can effectively differentiate between samples to some extent. Notably, the gradients of ED samples exhibit a more pronounced decay in this representation, due to the weak correlation with correlated ET gradients.

## 4.3 Adaptation in Tracked Low-Dimensional Subspace

The above findings offers a way to mitigate the impact of ED samples by leveraging the low-dimensional structure of ET gradients. We utilize the low-dimensional representation $\tilde{g}_t$ of the gradient $g_t$ for parameter update. Furthermore, considering that the test data distribution may vary continuously in continual adaptation scenarios, we dynamically track a low-dimensional principal subspace of the gradients in an online manner. As evidenced by experiments (Table 4 in Appendix), using a fixed subspace would degrade performance under varying test data distributions.

Specifically, during adaptation, we compute the gradient of the entropy loss with respect to $\theta$ for the $t$-th batch as $g_t = \nabla_\theta \mathcal{L}_e(\mathcal{B}_t; \theta)$. To capture the evolving gradient structure, we maintain a queue of the most recent $k$ gradients $G_t = [g_{t-k+1}, g_{t-k+2}, \dots, g_t] \in \mathbb{R}^{n \times k}$ where $n$ is the dimensionality of the parameter space. Subsequently, PCA (3) is applied to $G_t$ to extract a $r$-dimensional principal subspace represented by the projection matrix $U_{r,t} \in \mathbb{R}^{n \times r}$, where $r < k \ll n$.

The projection matrix $U_{r,t}$ defines the subspace used for parameter update during adaptation, based on which the component of $g_t$ residing within this subspace is obtained by projecting $g_t$ into the subspace and then back-projecting it into the original parameter space: $\tilde{g}_t = U_{r,t} U_{r,t}^\top g_t$. Then, the model parameters are updated based on $\tilde{g}_t$ as

$$\theta_{t+1} = \theta_t - \eta \tilde{g}_t,$$

where $\eta$ is the learning rate. This approach ensures that the parameter updates are informed by the salient components of ET gradients, captured by the dynamically tracked low-dimensional subspace.

However, applying this method to update all model parameters is memory-intensive, as it requires maintaining the gradient queue $G_t$ of size $n \times k$ and the projection matrix $U_{r,t}$ of size $n \times r$. This

significantly increases memory consumption, particularly for modern neural networks with large parameter dimensionality $n$, which makes it impractical for some on-device adaptation applications.

To address this issue, we restrict updates to the affine parameters of the normalization layers rather than the entire model. Previous studies [1, 2, 7] have demonstrated that updating normalization layers alone is sufficient to achieve strong performance in TTA, as they play a crucial role in controlling the feature distribution and are sensitive to distributional shifts. Notably, the BN and LN parameters typically constitute less than 1% of the total model parameters. Hence, tracking the subspace of them introduces minimal computational and memory overhead.

Moreover, we also employ the entropy-based sample filtering (ESF) strategy [15, 7] before subspace tracking. Note that, ESF alone cannot distinguish between ET and ED samples, as shown in Figure 7 in Appendix G. As shown in Section 4.1, ET sample gradients form a more prominent low-dimensional structure, while ED sample gradients are scattered and less-correlated. Thus, removing high-entropy (ambiguous) samples (Figure 7) enables the subspace to better capture reliable update directions, which makes ED gradients more orthogonal to it and easier to suppress.

## 5 Analysis on the Enhanced Stability of Subspace Projected TTA

We present stability analysis to show that the proposed subspace projection method enhances TTA robustness against detrimental ED samples. Note that TTA is essentially a local fine-tuning of a source pretrained model. The analysis is restricted to a neighbourhood of any local equilibrium, and relies on assumptions that ET gradients concentrate in a low-dimensional principal subspace, while ED gradients have weak inter-sample correlation and approximately isotropic dispersion.

Let $\theta_t \in \mathbb{R}^n$ be the model parameters updated by mini-batch SGD during TTA. For each step $t$, the mini-batch $\mathcal{B}_t = \{\hat{\mathcal{B}}_t, \breve{\mathcal{B}}_t\}$ contains ET samples $\hat{\mathcal{B}}_t$ and ED samples $\breve{\mathcal{B}}_t$. Denote the gradients on $\mathcal{B}_t$ as

$$g_t := \nabla_\theta L(\theta_t; \mathcal{B}_t) = \hat{g}_t + \breve{g}_t,$$

where $\hat{g}_t := \nabla_\theta L(\theta_t; \hat{\mathcal{B}}_t)$ and $\breve{g}_t := \nabla_\theta L(\theta_t; \breve{\mathcal{B}}_t)$ denote the gradient components of ET and ED samples, respectively. From the results in Section 4.1, we make assumptions: *a)* Mean and variance models of ET and ED gradients: $\mathbb{E}_\mathcal{B}[\hat{g}] = \bar{g}_{\mathrm{ET}}$, $\mathrm{Cov}[\hat{g}] = \Sigma_{\mathrm{ET}}$, $\mathbb{E}_\mathcal{B}[\breve{g}] = \bar{g}_{\mathrm{ED}}$, $\mathrm{Cov}[\breve{g}] = \Sigma_{\mathrm{ED}} = \sigma_{\mathrm{ED}}^2 I_n + \Delta$, where $\Sigma_{\mathrm{ET}}$ and $\Sigma_{\mathrm{ED}}$ are batch-normalized covariance of ET and ED gradients, respectively, with $\|\Delta\| \ll \sigma_{\mathrm{ED}}^2$. Weak correlation between ET and ED gradients: $\mathrm{Cov}(\hat{g} - \bar{g}_{\mathrm{ET}}, \breve{g} - \bar{g}_{\mathrm{ED}}) = 0$. *b)* Low–rank structure of ET gradients: $\Sigma_{\mathrm{ET}}$ has effective rank $r \ll n$, let $\Sigma_{\mathrm{ET}} = U\Lambda U^T$ be any eigen-decomposition with $U_r = [u_1, \ldots, u_r]$ being the eigenvectors corresponding to the largest eigenvalues, and denote $P_r := U_r U_r^T$. *c)* Small-step limit: As $\eta \to 0$ the discrete dynamics of SGD $\theta_{t+1} = \theta_t - \eta g_t$ converge to the stochastic differential equation (SDE) [52]

$$d\theta(t) = -\bar{g}(\theta)dt + \sqrt{\eta}\Sigma^{1/2}(\theta)dW(t), \tag{5}$$

where $\bar{g} = \bar{g}_{\mathrm{ET}} + \bar{g}_{\mathrm{ED}}$ and $\Sigma = \Sigma_{\mathrm{ET}} + \Sigma_{\mathrm{ED}}$.

**Theorem 1 (Local stability of full-space and subspace-projected TTA).** Suppose that the above assumptions hold. Let $\theta^\bullet$ be an equilibrium satisfying $\bar{g}(\theta^\bullet) = 0$ (i.e. with zero gradient expectation). For the full-space SGD, if, for all $\theta$ in a neighbourhood of $\theta^\bullet$,

$$\bar{g}_{\mathrm{ET}}^T \cdot (\theta - \theta^\bullet) > \left|\bar{g}_{\mathrm{ED}}^T \cdot (\theta - \theta^\bullet)\right| + \frac{\eta}{2}\mathrm{tr}(\Sigma), \tag{6}$$

then the SDE (5) is mean-square stable at $\theta^\bullet$. Further, for rank-$r$ subspace projection based learning, i.e., update in the principal subspace of ET gradients via replacing $g_t$ by $P_r g_t$, the corresponding continuous-time SDE

$$d\theta(t) = -P_r \bar{g}(\theta)dt + \sqrt{\eta}P_r \Sigma^{1/2}(\theta)dW(t), \tag{7}$$

is mean-square stable at $\theta^\bullet$ if

$$(P_r \bar{g}_{\mathrm{ET}})^T(\theta - \theta^\bullet) > \left|(P_r \bar{g}_{\mathrm{ED}})^T(\theta - \theta^\bullet)\right| + \frac{\eta}{2}\mathrm{tr}(P_r \Sigma P_r^T). \tag{8}$$

The proof is given in Appendix A. Theorem 1 provides balance-of-forces conditions for the stability of both full-space SGD and subspace-projected SGD. Stable local convergence is guaranteed when the corrective drift from ET gradients exceeds the combined ED bias and diffusion noise. Conversely,

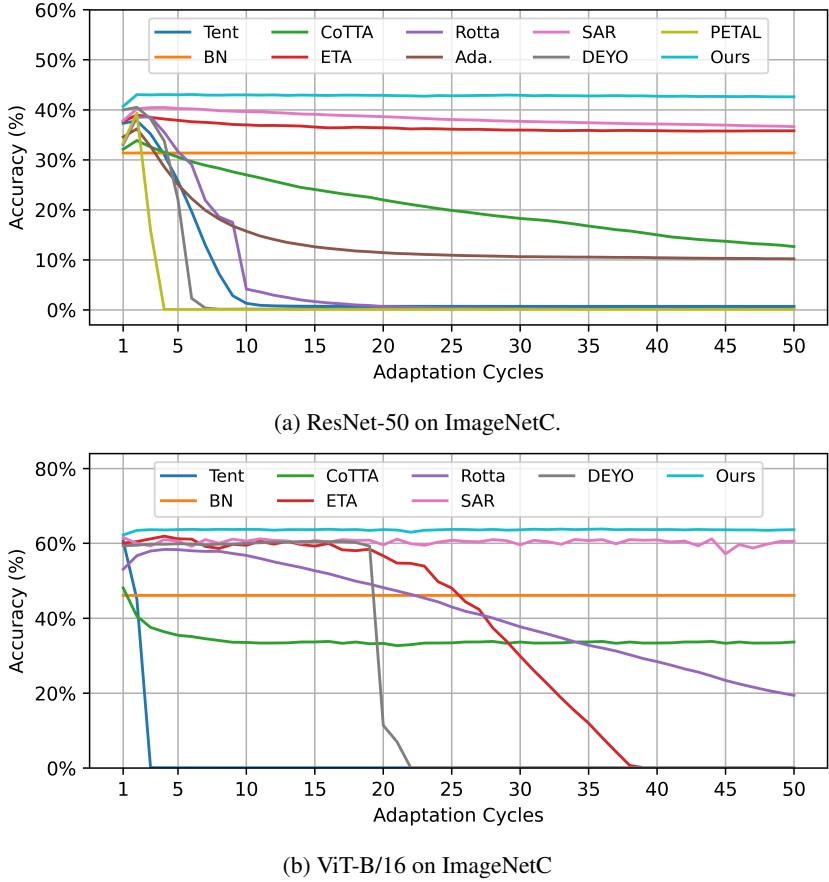

(a) ResNet-50 on ImageNetC.

(b) ViT-B/16 on ImageNetC

Figure 4: Accuracy of continual adaptation with ResNet-50 and ViT-B/16 over 50 cycles on ImageNetC. Each cycle contains 15 corruptions with 50000 samples for each corruption, resulting in a total of $3.75 \times 10^7$ samples used in the 50 cycles.

if the ED contribution outweighs corrective ET drift, the dynamics may diverge or converge to an inferior minimum, which leads to model degeneration. Comparing the conditions (6) and (8), the subspace projection method relaxes the stability requirement and thus is more robust.

Specifically, when the ET gradient mean $\bar{g}_{ET}$ lies mostly in the subspace $U_r$, we have $P_r \bar{g}_{ET} \approx \bar{g}_{ET}$. Meanwhile, projection onto this subspace removes most out-of-subspace component of the ED-gradient bias such that $P_r \bar{g}_{ED} \approx 0$ for $r \ll n$. Then, the stability condition (8) simplifies to

$$\bar{g}_{ET}^T \cdot (\theta - \theta^{\bullet}) > \frac{\eta}{2}\Big(\sum_{i=1}^{r} \lambda_i + r\sigma_{ED}^2\Big), \tag{9}$$

where $\sum_{i=1}^{r} \lambda_i = \mathrm{Tr}(P_r \Sigma_{ET} P_r^T)$ with $\lambda_i$ being the $i$-th eigenvalue of $\Sigma_{ET}$. Compared with the full-space case with $\mathrm{tr}(\Sigma) = \sum_{i=1}^{n} \lambda_i + n\sigma_{ED}^2$, subspace projection largely reduces the diffusion energy by a factor $\frac{\sum_{i=1}^{r} \lambda_i + r\sigma_{ED}^2}{\sum_{i=1}^{n} \lambda_i + n\sigma_{ED}^2} \approx \frac{r}{n} \ll 1$. This significantly enlarges the admissible range of ED bias and noise that still satisfies the stability condition. Consequently, the subspace-projected method is substantially more robust to detrimental ED samples.

## 6 Experiments

We conduct experiments on the ImageNetC dataset [53], which consists of 15 corruptions each with 5 severity levels. We experiment with two representative model architectures: ResNet-50 with batch normalization [54] and the ViT-B/16 with layer normalization. Results on CIFAR100C and semantic segmentation are provided in Appendix K and L.

We evaluate our method under a challenging long-term continual TTA setting, where the model adapts continuously over 50 cycles of 15 corruption types (*severity=5*), a total of 37.5 million test samples. The model performs unsupervised continual adaptation without any external intervention from the very first beginning, such as domain-specific information, model resetting, or warm-up. We compare our method ($r = 25$) with several SOTA continual TTA methods, including AdaContrast [5], BN [1], TENT [2], CoTTA [8], SAR [7], RoTTA [4], ETA [15], PETAL [55] and DeYO [56]. Notably, CoTTA, and RoTTA employ teacher-student networks, while SAR adopts a model resetting strategy.

## 6.1 Results on ImageNet-to-ImageNetC

Figure 4 presents the results on the ImageNet-to-ImageNetC task over 50 continual adaptation cycles. Clearly, our method demonstrates robust and superior long-term adaptation performance on ImageNetC. Most of the compared methods suffer significant degradation during long-term adaptation. For instance, the accuracy of Tent drops below 10% within the first 10 adaptation cycles, whilst that of CoTTA gradually declines over continual adaptation. In contrast, our method consistently achieves high performance throughout the entire adaptation process, attributed to its ability to effectively suppress the detrimental impact of ED samples. These results demonstrate the robustness and adaptability of our approach in challenging long-term adaptation scenarios.

Results under a standard TTA setting, where a model adapts to one corruption at a time for a single epoch, are provided in Tables 2 and 3 in Appendix C, which demonstrate that our method can also achieve competitive performance in short-term adaptation scenarios.

## 6.2 Analysis and Ablations

**Ablation of subspace projection.** We evaluate the effectiveness of the subspace projection method. As shown in Table 1, the method without using any strategy quickly collapse under entropy-based continual adaptation. Entropy filtering improves initial performance but still suffers from ED samples due to the nature of entropy, leading to degeneration in long-term. Our method achieves consistent robustness over long-term adaptation.

Table 1: Accuracy (%) at different adaptation cycles on ImageNetC with ResNet-50.

| Strategy | Cycle 1 | Cycle 25 | Cycle 50 | Cycle 75 | Cycle 100 |
|---|---|---|---|---|---|
| / | 39.13 | 0.70 | 0.69 | 0.70 | 0.70 |
| +Entropy filtering | 37.70 | 36.09 | 34.79 | 31.56 | 28.68 |
| +Subspace | 36.71 | 36.45 | 36.87 | 36.34 | 36.68 |
| +Entropy filtering and Subspace (Ours) | 40.70 | 42.70 | 43.10 | 42.79 | 42.66 |

**Effect of subspace dimension.** Figure 5 shows the performance of our method with different dimensions of the subspace, $r \in \{10, 25, 50, 100\}$, in continual adaptation on ImagenetC over 50 cycles. Using a reasonably low dimension (e.g., $r = 25$) can achieve satisfactory performance during long-term continual adaptation. However, as the rank increases, the restricting effect of the subspace diminishes, leading to performance degradation in later cycles. On the other hand, too low a rank would limit the adaptation performance.

**Effect of hyperparameters.** We conduct experiments on the sampling interval and the length of the gradient queue $k$. As shown in Figure 6, when the queue length $k$ for storing gradients exceeds a certain threshold (e.g., 25), it captures sufficient gradient information to compute a stable subspace, thus maintaining stable performance. Similarly, the model exhibits robust long-term adaptation performance across a wide range of sampling intervals. Table 5 in Appendix E further shows that proposed subspace method exhibits strong robustness to a wide range of learning rate choices.

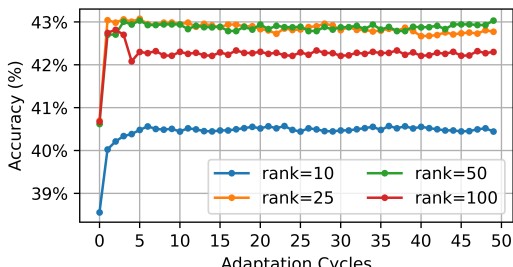

Figure 5: Ablation on subspace dimentsion on ImageNetC with ResNet-50.

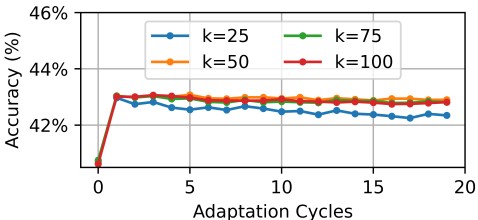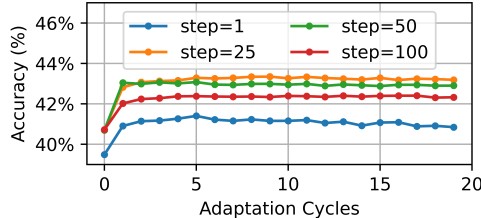

Figure 6: Results on gradient queue length $k$ (left) and sampling interval(right).

**Ablation on subspace tracking.** Ablation study on the proposed subspace tracking approach is provided in Table 4 in Appendix D, which demonstrates the effectiveness and indispensability of it in achieving robust continual adaptation under continually varying data distributions.

**Computational Complexity and Efficiency.** Our method only introduces additional memory to store a gradient queue to compute subspace (Section 4.3), which is typically less than $0.01kn \approx 0.5n$ for $k = 50$. Thus, it does not incur significant memory and computational costs. Memory and runtime comparison on ImageNetC with ResNet-50 is given in Table 7 in Appendix I.

## 7 Conclusion

To address the critical challenge of performance degeneration in unsupervised continual TTA, this work identified ED samples as an underlying cause of the degeneration in entropy-minimization methods. Furthermore, we revealed that the backward gradients of entropy-minimization exhibit an intrinsic low-dimensional structure, and demonstrated that constraining weight updates within a low-dimensional principal subspace can effectively suppress the detrimental impact of ED samples. Then, we proposed a novel subspace-based continual TTA method and proved its enhanced robustness against detrimental ED samples. Extensive experiments demonstrated that LCoTTA can effectively overcome degeneration and maintain robust performance in long-term adaptation scenarios.

**Limitations.** In this work, we only consider a fixed subspace dimension $r$, while an adaptive selection of $r$ may further improve performance, e.g., larger at the beginning and lower later, or adjusting it based on the degree of distribution shift and severity changes. Moreover, our theoretical analysis of the stability of the subspace-based method holds only locally.

## 8 Acknowledgment

This work was supported in part by the Science and Technology Innovation 2030-Major Project under Grant 2022ZD0208701, and National Natural Science Foundation of China (NSFC) under Grant 62271314. The authors also express special thanks to *Yepeng Yang*, for the helpful discussion related to this work.

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

# A  Proof of Theorem 1

Consider the SGD update rule employed during TTA as

$$\theta_{t+1} = \theta_t - \eta g(\theta_t; \mathcal{B}_t), \tag{10}$$

where $g(\theta_t; \mathcal{B}_t) = \nabla_\theta L(\theta_t; \mathcal{B}_t)$ is gradient on the batch $\mathcal{B}_t$. Assume the batch $\mathcal{B}_t$ contains both ET and ED samples, where the ET sample subset is denoted by $\hat{\mathcal{B}}_t$ whilst the ED sample subset is denoted by $\breve{\mathcal{B}}_t$ such that $\mathcal{B}_t = [\hat{\mathcal{B}}_t, \breve{\mathcal{B}}_t]$. Denote the gradients on $\hat{\mathcal{B}}_t$ and $\breve{\mathcal{B}}_t$ are given by

$$\hat{g}_t := g(\theta_t; \hat{\mathcal{B}}_t) = \nabla_\theta L(\theta_t; \hat{\mathcal{B}}_t),$$

$$\breve{g}_t := g(\theta_t; \breve{\mathcal{B}}_t) = \nabla_\theta L(\theta_t; \breve{\mathcal{B}}_t).$$

Denote the mean and covariance of ET and ED gradients by

$$\mathbb{E}_\mathcal{B}[\hat{g}] = \bar{g}_{\text{ET}}, \quad \text{Cov}_\mathcal{B}(\hat{g}) = \Sigma_{\text{ET}},$$

$$\mathbb{E}_\mathcal{B}[\breve{g}] = \bar{g}_{\text{ED}}, \quad \text{Cov}_\mathcal{B}(\breve{g}) = \Sigma_{\text{ED}},$$

where $\Sigma_{\text{ET}}$ and $\Sigma_{\text{ED}}$ are the batch-normalized covariance of ET and ED gradients, respectively. Empirical results in Section 4.1 indicate weak inter-sample correlation among ED-sample gradients. We therefore model

$$\nabla_\theta L(\theta_t; \breve{\mathcal{B}}_t) = \bar{g}_{\text{ED}} + \xi_{\text{ED}}, \tag{11}$$

with $\mathbb{E}[\xi_{\text{ED}}] = 0$, and $\text{Cov}(\xi_{\text{ED}}) = \Sigma_{\text{ED}}$, where $\bar{g}_{\text{ED}}$ is the bias of ED gradients. $\xi_{\text{ED}}$ is scattered noise with covariance $\Sigma_{\text{ED}} = \sigma_{\text{ED}}^2 I_n + \Delta$ with $\|\Delta\| \ll \sigma_{\text{ED}}^2$.

TTA operates as a local fine-tuning of a pretrained model. The expected ET gradient $\bar{g}_{\text{ET}}$ points toward a nearby local minimum, whereas its covariance $\Sigma_{\text{ET}}$ is low-rank (Figure 2), with effective rank $r \ll n$. In contrast, the ED gradient dispersion is almost isotropic, $\Sigma_{\text{ED}} \approx \sigma_{\text{ED}}^2 I_n$. Moreover, assuming $\mathbb{E}_\mathcal{B}[(\hat{g} - \bar{g}_{\text{ET}})(\breve{g} - \bar{g}_{\text{ED}})^T] = 0$, the mean and covariance of the mini-batch gradient $g(\theta_t; \mathcal{B}_t)$ can be expressed as

$$\mathbb{E}_\mathcal{B}[g] = \bar{g}_{\text{ET}} + \bar{g}_{\text{ED}}, \quad \text{Cov}_\mathcal{B}(g) = \Sigma_{\text{ET}} + \Sigma_{\text{ED}}$$

When the learning rate $\eta \to 0$, the discrete dynamics (10) can be approximated by the continuous-time stochastic differential equation (SDE) [52]

$$d\theta(t) = -\bar{g}(\theta)dt + \sqrt{\eta}\Sigma(\theta)^{1/2}dW(t). \tag{12}$$

with drift $\bar{g}(\theta) = \bar{g}_{\text{ET}} + \bar{g}_{\text{ED}}$ and diffusion $\Sigma(\theta) = \Sigma_{\text{ET}} + \Sigma_{\text{ED}}$.

Define a local expected-gradient equilibrium $\theta^\bullet$ that

$$\bar{g}(\theta^\bullet) = \bar{g}_{\text{ET}}(\theta^\bullet) + \bar{g}_{\text{ED}}(\theta^\bullet) = 0, \tag{13}$$

at which ET corrective force balances ED bias. Choose the quadratic Lyapunov function $V(\theta) = \frac{1}{2}\|\theta - \theta^\bullet\|^2$, its infinitesimal generator is

$$\mathcal{L}_V = -\bar{g}(\theta)^T \cdot (\theta - \theta^\bullet) + \frac{\eta}{2}\text{Tr}(\Sigma(\theta)). \tag{14}$$

Then, a sufficient condition for mean-square stability is given by

$$\bar{g}_{\text{ET}}^T \cdot (\theta - \theta^\bullet) > -\bar{g}_{\text{ED}}^T \cdot (\theta - \theta^\bullet) + \frac{\eta}{2}\text{Tr}(\Sigma). \tag{15}$$

In the worst case that the ED bias is opposite to $(\theta - \theta^\bullet)$, it follows from (15) that

$$\bar{g}_{\text{ET}}^T \cdot (\theta - \theta^\bullet) > \left|\bar{g}_{\text{ED}}^T \cdot (\theta - \theta^\bullet)\right| + \frac{\eta}{2}\text{Tr}(\Sigma). \tag{16}$$

If either the bias magnitude $\|\bar{g}_{\text{ED}}\|$ or the noise level $\text{Tr}(\Sigma)$ is too large that exceeds a critical threshold, stable convergence cannot be guaranteed.

Next, we consider subspace projection based SGD to show its advantage. Retain only the first principal ET directions $U_r$ and let $P_r = U_r U_r^T$ be the projection matrix. With $P_r \Sigma P_r^T = P_r \Sigma^{1/2}(P_r \Sigma^{1/2})^T$, the subspace projected SDE can be expressed as

$$d\theta(t) = -P_r \bar{g}(\theta)dt + \sqrt{\eta}P_r \Sigma(\theta)^{1/2}dW(t). \tag{17}$$

Similarly, we can derive the corresponding stability condition as

$$(P_r \bar{g}_{\text{ET}})^T \cdot (\theta - \theta^\bullet) > \left|(P_r \bar{g}_{\text{ED}})^T \cdot (\theta - \theta^\bullet)\right| + \frac{\eta}{2} \text{Tr}(P_r \Sigma P_r^T). \tag{18}$$

As ET directions lie almost entirely in $U_r$, we have $P_r \bar{g}_{\text{ET}} \approx \bar{g}_{\text{ET}}$. Meanwhile, the subspace projection eliminates most ED bias outside the subspace, hence for $r \ll n$ we have $P_r \bar{g}_{\text{ED}} \approx 0$. Moreover, diffusion noise is greatly reduced by subspace projection as

$$\text{Tr}(P_r \Sigma P_r^T) \approx \sum_{i=1}^{r} \lambda_i + r\sigma_{\text{ED}}^2, \tag{19}$$

where $\lambda_i$ is the $i$-th eigenvalue of $\Sigma_{\text{ET}}$, with $\text{Tr}(P_r \Sigma_{\text{ET}} P_r^T) = \sum_{i=1}^{r} \lambda_i$. Then, under these assumptions and with $r \ll n$, the sufficient condition (18) can be simplified as

$$(\bar{g}_{\text{ET}})^T \cdot (\theta - \theta^\bullet) > \left|(P_r \bar{g}_{\text{ED}})^T \cdot (\theta - \theta^\bullet)\right| + \frac{\eta}{2} \text{Tr}(P_r \Sigma P_r^T)$$
$$\approx \frac{\eta}{2} \left( \sum_{i=1}^{r} \lambda_i + r\sigma_{\text{ED}}^2 \right). \tag{20}$$

Compared with the condition (16) for full-space SGD, the noise term $\text{Tr}(\Sigma) = \sum_{i=1}^{n} \lambda_i + n\sigma_{\text{ED}}^2$ is largely reduced to $\sum_{i=1}^{r} \lambda_i + r\sigma_{\text{ED}}^2$. In the setting of the proposed subspace projection method with $r \ll n$, we have

$$\sum_{i=1}^{r} \lambda_i + r\sigma_{\text{ED}}^2 \ll \sum_{i=1}^{n} \lambda_i + n\sigma_{\text{ED}}^2. \tag{21}$$

Consequently, it is easy to see that, the projection onto the subspace $U_r$ can substantially enhance the robustness of entropy minimization based TTA against the detrimental effect of ED samples.

## B  Updating in Low-Dimensional Subspaces Constrains the Adaptation

Adapting model parameters within a low-dimensional subspace offers a controlled mechanism for constraining weight changes during adaptation, which in turn enhances the stability of the process. Let $\theta_s \in \mathbb{R}^n$ represent the parameter vector of the pre-trained source model, and $U_r \in \mathbb{R}^{n \times r}$ denote an orthonormal basis spanning a $r$-dimensional subspace, where $r \ll n$. Weight updates with and without subspace constraints can be compared as follows.

When the updates are restricted to the subspace defined by $U_r$, the updated weights are expressed as $\tilde{\theta}_t = \theta_s + U_r U_r^T \Delta\theta$, whereas without the subspace constraint, the updates are given by $\theta_t = \theta_s + \Delta\theta$. Let $U = [U_r, U_r^\perp] \in \mathbb{R}^{n \times n}$ represent the full orthonormal basis spanning the original parameter space, where $U_r \in \mathbb{R}^{n \times r}$ and $U_r^\perp \in \mathbb{R}^{n \times (n-r)}$ span two orthogonal complementary subspaces. Denote

$$v_s = U_r^T \theta_s \in \mathbb{R}^r, \quad \breve{v}_s = {U_r^\perp}^T \theta_s \in \mathbb{R}^{n-r},$$
$$\delta v = U_r^T \Delta\theta \in \mathbb{R}^r, \quad \delta\breve{v} = {U_r^\perp}^T \Delta\theta \in \mathbb{R}^{n-r}.$$

Using the orthonormality of $U$, the weight updates with and without subspace representations can be expressed as

$$\tilde{\theta}_t = \theta_s + U_r U_r^T \Delta\theta = U \begin{bmatrix} v_s + \delta v \\ \breve{v}_s \end{bmatrix},$$

$$\theta_t = \theta_s + \Delta\theta = U \begin{bmatrix} v_s + \delta v \\ \breve{v}_s + \delta\breve{v} \end{bmatrix}.$$

Clearly, in the space defined by $U$, the update $\tilde{\theta}_t$ involves only a much smaller $r$-dimensional subspace, as $r \ll n$, in contrast to the full update $\theta_t$. In our experiments, we use $r = 5$, resulting in an extremely low-dimensional subspace for adaptation.

Given that $U_r$ has orthonormal columns, satisfying $U_r^T U_r = I_{r \times r}$, for any $\Delta\theta \in \mathbb{R}^n$, we have $\|U_r U_r^T \Delta\theta\|_2^2 = (\Delta\theta^T U_r)(U_r^T \Delta\theta) = \|U_r^T \Delta\theta\|_2^2 \leq \|\Delta\theta\|_2^2$. This implies that the magnitude of the weight updates is reduced when using subspace projection as

$$\|U_r U_r^T \Delta\theta\|_2 \leq \|\Delta\theta\|_2.$$

Thus, adapting weights within a low-dimensional subspace reduces the magnitude of changes and ensures that the updated weights retain higher dependency on the source model. This property, together with the suppression effect of low-dimensional principal subspace on deceptive samples (Section 4.2), benefits the stability in long-term continual adaptation, with improved robustness to hyperparameters (see Table 5 in the ablation study).

## C  Results of Single-Epoch Adaptation on ImageNetC

Table 2 and Table 3 present the performance under the standard TTA setting over a single cycle, demonstrating that our method can still achieve competitive results in short-term adaptation scenarios. We also report the performance of several more recent methods including BeCoTTA [26], AEA [57], and TCA [58]. Since some of these methods do not have publicly available code for reproduction, we directly cite their reported results from their respective papers for comparison.

Table 2: Comparison of accuracy (%) over a single cycle of 15 corruptions on the ImageNetC dataset with ResNet-50. [†]The reported results of BeCoTTA [26], AEA [57], and TCA [58] are directly taken from their respective original papers.

| | | | | | | | | | | | | | | | | $t \longrightarrow$ |
| Method | Gaussian | Shot Noise | Impulse | Defocus | Glass | Motion | Zoom | Snow | Frost | Fog | Brightness | Contrast | Elastic | Pixelate | JPEG | Mean |
|---|---|---|---|---|---|---|---|---|---|---|---|---|---|---|---|---|
| Source | 2.2 | 2.9 | 1.8 | 18.3 | 10.2 | 14.8 | 22.1 | 16.5 | 22.9 | 24.1 | 58.7 | 5.5 | 17.5 | 20.7 | 31.4 | 18.0 |
| CoTTA | 15.4 | 18.1 | 19.4 | 18.4 | 20.9 | 31.5 | 41.7 | 39.4 | 38.5 | 51.8 | 63.6 | 33.2 | 52.9 | **59.4** | 54.8 | 37.3 |
| SAR | 17.8 | 25.5 | 28.1 | 22.7 | 26.1 | 33.7 | 43.5 | 38.4 | 36.9 | 49.2 | 62.7 | 30.2 | 49.9 | 54.1 | 48.8 | 37.8 |
| RoTTA | 11.9 | 17.4 | 17.4 | 9.5 | 16.1 | 26.8 | 39.7 | 34.1 | 35.3 | 46.2 | 64.5 | 25.3 | 45.2 | 52.2 | 47.3 | 32.6 |
| AdaCon | 17.1 | 19.4 | 21.1 | 18.0 | 22.2 | 26.3 | 36.3 | 37.5 | 36.3 | 47.2 | 61.6 | 33.0 | 45.1 | 50.9 | 46.8 | 34.6 |
| DeYO | 25.8 | **34.9** | **35.7** | 26.4 | **30.7** | 35.3 | 43.3 | 37.3 | 36.7 | 47.9 | 60.4 | 33.5 | 48.7 | 53.3 | 50.0 | 40.0 |
| PETAL | 12.9 | 14.5 | 15.7 | 13.6 | 16.4 | 26.4 | 37.5 | 36.8 | 36.0 | 47.6 | 61.5 | 26.1 | 49.9 | 55.7 | 49.5 | 33.3 |
| ETA | 24.4 | 34.2 | 34.9 | 26.2 | 30.3 | 32.6 | 38.4 | 34.2 | 34.2 | 42.8 | 54.6 | 34.5 | 46.2 | 49.8 | 47.7 | 37.7 |
| BeCoTTA[†] | 15.9 | 25.7 | 27.8 | 22.6 | 28.1 | 36.6 | 44.9 | 42.8 | 38.8 | 49.3 | 63.6 | 33.9 | 50.8 | 54.4 | 51.6 | 39.1 |
| AEA[†] | **26.2** | 26.8 | 27.3 | 24.2 | 20.8 | **40.3** | **48.1** | **47.3** | **41.4** | **56.0** | **65.7** | 9.5 | **53.4** | 56.7 | 49.5 | 39.5 |
| TCA[†] | 21.7 | 28.2 | 26.5 | 25.6 | 26.5 | 36.7 | 43.5 | 43.1 | 40.6 | 51.9 | 60.4 | **40.4** | 52.8 | 57.1 | **55.3** | **40.7** |
| Ours | 24.3 | 33.7 | 34.3 | 26.1 | 30.4 | 37.0 | 44.7 | 38.7 | 39.6 | 48.9 | 61.3 | 35.7 | 50.9 | 54.2 | 50.7 | **40.7** |

Table 3: Comparison of accuracy (%) over a single cycle of 15 corruptions on the ImageNetC dataset with ViT-B/16.

| | | | | | | | | | | | | | | | | $t \longrightarrow$ |
| Method | Gaussian | Shot Noise | Impulse | Defocus | Glass | Motion | Zoom | Snow | Frost | Fog | Brightness | Contrast | Elastic | Pixelate | JPEG | Mean |
|---|---|---|---|---|---|---|---|---|---|---|---|---|---|---|---|---|
| Tent | 59.8 | 63.4 | 62.9 | 45.5 | 50.1 | 58.6 | 50.9 | 63.3 | 60.5 | **66.5** | **78.6** | 55.4 | 54.6 | 69.7 | 70.2 | 60.7 |
| BN | 50.1 | 51.0 | 51.1 | 31.5 | 27.6 | 44.1 | 39.5 | 52.5 | 47.7 | 44.2 | 75.2 | 8.9 | 44.3 | 60.9 | 63.3 | 46.1 |
| CoTTA | 59.9 | 62.9 | 62.7 | 44.4 | 48.3 | 55.6 | 47.4 | 61.2 | **65.5** | 52.2 | 74.3 | 23.5 | **67.5** | **72.5** | **71.5** | 58.0 |
| ETA | 59.5 | **63.7** | **63.2** | 52.3 | 52.4 | 58.6 | 55.6 | 64.6 | 62.5 | 63.7 | 77.9 | 50.3 | 59.4 | 70.1 | 71.4 | 61.7 |
| RoTTA | 57.7 | 60.0 | 60.4 | 41.9 | 34.3 | 51.5 | 43.5 | 64.7 | 63.4 | 35.8 | 78.2 | 21.7 | 47.7 | 67.0 | 68.1 | 53.1 |
| SAR | 59.1 | 61.2 | 61.5 | **54.2** | 55.3 | 58.5 | **55.8** | 60.9 | 61.9 | 64.6 | 76.8 | **58.3** | 58.2 | 68.4 | 68.9 | 61.6 |
| DeYO | 59.2 | 61.6 | 60.8 | 44.6 | 47.9 | 56.8 | 49.3 | 61.8 | 61.6 | 62.8 | 77.0 | 56.3 | 54.6 | 66.2 | 69.7 | 59.4 |
| Ours | **60.6** | 62.8 | 62.5 | 52.2 | **55.4** | **58.9** | 54.5 | **65.3** | 63.4 | 66.2 | **78.6** | 54.5 | 60.7 | 69.2 | 69.2 | **62.3** |

## D  Effectiveness of the Proposed Subspace Tracking Approach

We conduct ablation experiments to evaluate the effect of the proposed subspace tracking approach on continual adaptation on ImageNetC with ResNet-50. We consider four variants: *1)* our method without using subspace projection of the gradients; *2)* our method using a fixed 5-dimensional subspace extracted from source data, denoted as "from source domain"; *3)* our method using a 5-dimensional subspace tracked online but delayed for one corruption type, denoted as "tracked with delay"; *4)* our method using the proposed subspace tracking approach, denoted as "tracked online". Table 4 shows the results of these variants for continual adaptation over one-cycle of 15 corruptions,

which demonstrates the effectiveness and indispensability of the proposed dynamic subspace tacking approach in achieving robust continual adaptation.

Table 4: Ablation on the effectiveness of the subspace tracking on ImageNetC with ResNet-50.

| Subspace | Subspace Type | ACC (%) |
|----------|---------------|---------|
| × | - | 18.0 |
| ✓ | From Source Domain | 27.0 |
| ✓ | Tracked with delay | 24.5 |
| ✓ | Tracked online | 40.7 |

## E    Robustness to Learning Rate

Table 5 presents a comparison between Tent and our method across different learning rates in continual adaptation on ImageNet-C. Consistent with results from previous studies [2, 8], entropy-minimization-based TTA methods, such as Tent, are highly sensitive to the choice of learning rate. In contrast, our method is much more robust to variation in learning rate, attributed to its ability to suppress deceptive samples and constrain updates within a principal subspace. This robustness substantially alleviates the need for meticulous hyperparameter tuning, offering a more practical and reliable solution for continual adaptation.

Table 5: ACC (%) of Tent and our method for different learning rates (LR) in continual adaptation on ImageNetC over one cycle with ResNet-50.

| Method \ LR | 0.0001 | 0.001 | 0.002 | 0.005 |
|-------------|--------|-------|-------|-------|
| Tent | 36.71 | 30.84 | 11.33 | 4.51 |
| Subpsace projection | 40.36 | 39.87 | 38.82 | 30.85 |

## F    Standard Deviation of Our Method

Table 6 presents the standard deviations of our method with different random seeds on CIFAR100-C with ResNext-29, ImageNetC with ResNet-50, and ImageNetC with ViT-B/16. We use five different random seeds and report the mean and standard deviation.

Table 6: Standard deviation (%) over 5 different random seeds on various datasets and models.

| Dataset & Model | Cycle 1 | Cycle 10 | Cycle 20 | Cycle 30 | Cycle 40 | Cycle 50 |
|-----------------|---------|----------|----------|----------|----------|----------|
| ResNeXt-29 on Cifar100 | $67.85 \pm 0.30$ | $68.24 \pm 0.27$ | $68.22 \pm 0.16$ | $68.07 \pm 0.40$ | $67.98 \pm 0.09$ | $68.09 \pm 0.19$ |
| ResNet-50 on ImagenetC | $40.70 \pm 0.42$ | $42.95 \pm 0.25$ | $42.83 \pm 0.34$ | $42.81 \pm 0.11$ | $42.70 \pm 0.21$ | $42.60 \pm 0.29$ |
| ViT-B/16 on ImagenetC | $62.25 \pm 0.17$ | $63.72 \pm 0.18$ | $63.56 \pm 0.05$ | $63.78 \pm 0.43$ | $63.70 \pm 0.28$ | $63.66 \pm 0.12$ |

## G    Sample Entropy Visualization of Model Prediction

Figure 7 shows the distribution of prediction entropy from the network when the test inputs are sorted by ERS. It can be observed that **a naive entropy filtering strategy can remove samples with high entropy values in the middle range, thereby robustifying adaptation performance. However, since many entropy-deceptive (ED) samples can also exhibit low entropy values, they cannot be filtered out by simple entropy filtering strategy**. This phenomenon is more pronounced on the more challenging ImageNet dataset, which explains why the filtering strategy still fails in long-term adaptation scenarios.

It can be seen from Figure 7 that, using entropy-based sample filtering (ESF) alone cannot distinguish between ET and ED samples. Thus, using ESF alone still suffers from degeneration in continual TTA.

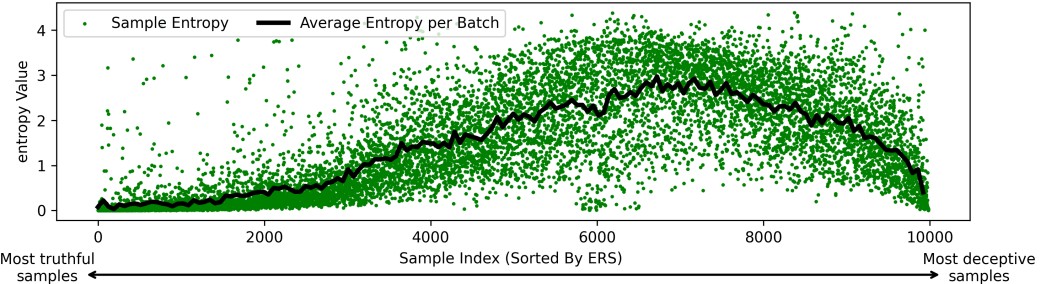

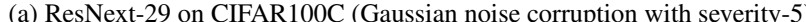

(a) ResNext-29 on CIFAR100C (Gaussian noise corruption with severity-5)

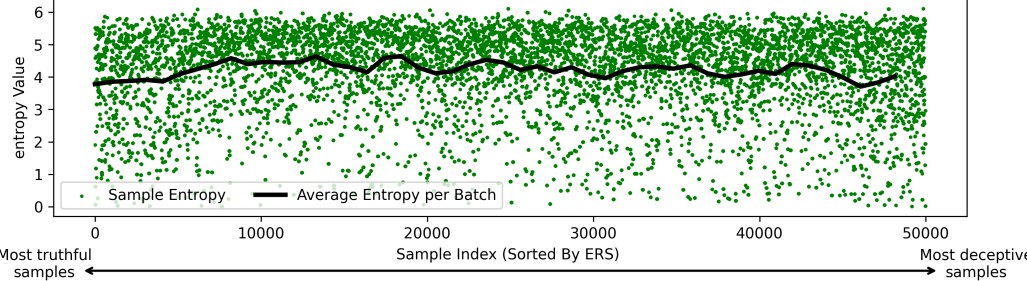

(b) ResNet-50 on ImageNetC (Gaussian noise corruption with severity-5)

Figure 7: Scatter plot of the sample entropy of model predictions with samples sorted by ERS. (a) The samples are from the Gaussian noise corruption of CIFAR100C with severity level 5. The predictions are obtained from a pretrained ResNext-29 model. (b) The samples are from the Gaussian noise corruption of ImageNetC with severity level 5. The predictions are obtained from a pretrained ResNet-50 model.

## H   The Reduction of Entropy-Deceptive Samples During Continual Adaptation

We conduct experiment to record the number changes of ED samples during the adaptation process. As shown in Figure 8, during continual adaptation, the number of ED samples gradually reduces.

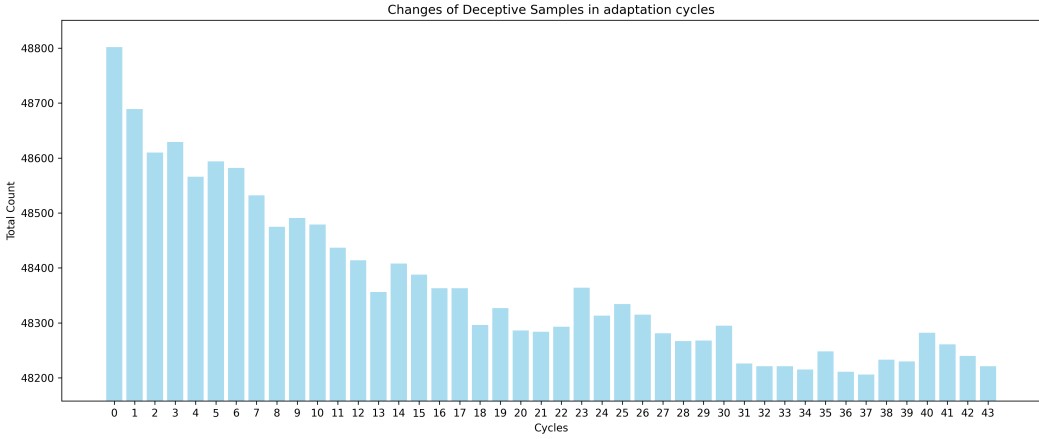

Figure 8: Decrease of entropy-deceptive (ED) samples during the adaptation process with ResNet-50 on the ImageNetC dataset.

# I    Comparison of Computational Complexity and Efficiency

As discussed in Section 6.2, our method only introduces an additional gradient queue and negligible PCA computation overhead. Table 7 presents a comparison of memory consumption and runtime among different methods. As can be seen, the computational and time complexity of our method are comparable to those of Tent, and our approach remains competitive among the compared methods.

Table 7: Runtime and memory comparison for one-cycle on ImageNetC with ResNet-50.

| Methods / Metrics | CoTTA | RoTTA | ETA | SAR | Ada | Tent | Ours |
|---|---|---|---|---|---|---|---|
| Runtime (s) | 659.9 | 894.4 | 232.3 | 355.7 | 1297 | 241.1 | 257.6 |
| Memory (GB) | 10.86 | 15.47 | 10.37 | 10.37 | 12.22 | 10.37 | 10.67 |

# J    Low-Dimension Structure of Gradients for ED and ET Samples

As discussed in Section 4.1, ET sample gradients are highly correlated and align well in parameter space, forming a clear low-dimensional structure. In contrast, ED gradients are scattered and lack such alignment. To illustrate this, we sort samples by ERS and analyze the top, middle, and last 20% using PCA. As shown in Figure 9, only ET samples reveal a pronounced low-dimensional subspace.

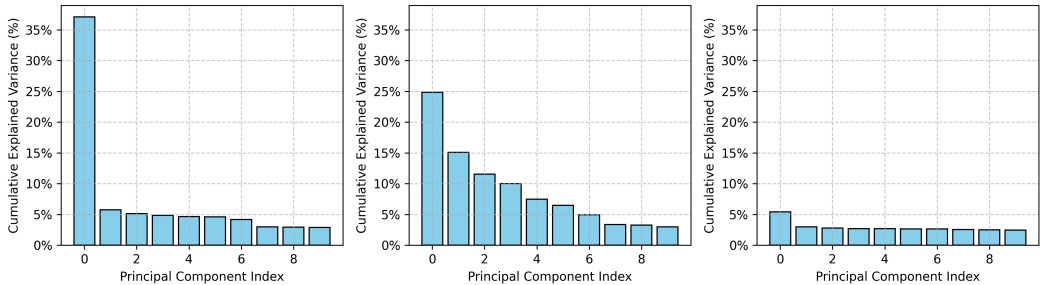

Figure 9: PCA visualization of sample gradients under different ERS rankings. **Left:** Top 20% ERS samples (**all ET samples**). **Middle:** Middle 20% ERS samples (**mixture of ET and ED samples**). **Right:** Last 20% ERS samples (**all ED samples**). The results illustrate that top ERS samples (ET samples) exhibit a more pronounced low-dimensional structure.

# K    Results on CIFAR100-to- CIFAR100C

## K.1    Results of Continual Adaptation on CIFAR100C

Figure 10 presents the performance comparison in the considered long-term adaptation setting on the CIFAR100-to- CIFAR100C task over 100 adaptation cycles. Using teacher-student networks and partial model resetting, CoTTA and ECoTTA mitigate the degeneration but still degrade over time due to the inherent instability of unsupervised adaptation. AdaContrast achieves strong initial performance but also degenerates over prolonged adaptation. In contrast, our method maintains robust performance throughout all adaptation cycles, effectively overcoming the degeneration issue in long-term continual adaptation.

## K.2    Results of Single Epoch on CIFAR100C

Table 8 presents the results of single epoch adaptation on CIFAR100C with ResNeXt-29.

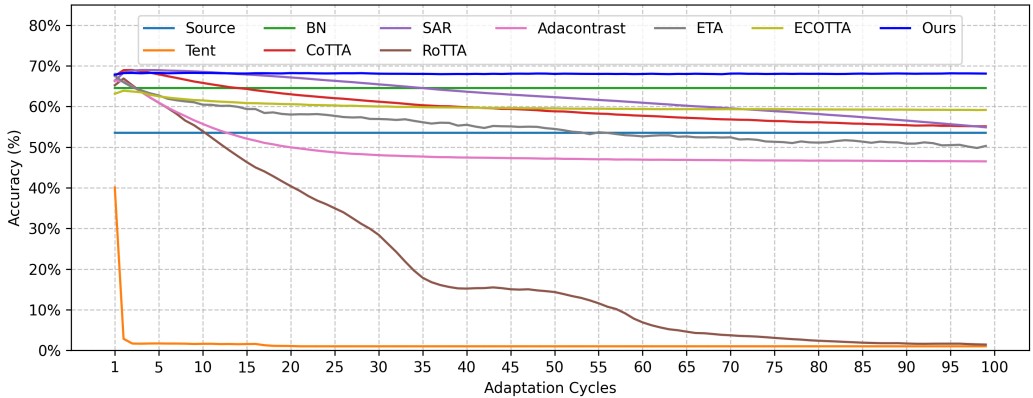

Figure 10: Accuracy of continual adaptation with ResNext-29 over 100 cycles on Cifar100C. Each cycle contains 15 corruptions with 10000 samples for each corruption, resulting in a total of $1.5 \times 10^7$ samples used in the 100 cycles.

Table 8: Accuracy (%) comparison of the methods over a single cycle on the CIFAR100C dataset with ResNeXt-29.

| | | | | | | | | | | | | | | | | $t \longrightarrow$ |
| Method | Gaussian | Shot Noise | Impulse | Defocus | Glass | Motion | Zoom | Snow | Frost | Fog | Brightness | Contrast | Elastic | Pixelate | JPEG | Mean |
|---|---|---|---|---|---|---|---|---|---|---|---|---|---|---|---|---|
| Tent | 62.9 | 64.3 | 58.5 | 62.6 | 49.1 | 51.9 | 51.4 | 41.3 | 36.5 | 29.8 | 30.2 | 19.5 | 15.8 | 15.3 | 12.1 | 40.1 |
| BN | 57.7 | 59.2 | 56.7 | 72.3 | 58.1 | 70.2 | 72.1 | 64.9 | 65.0 | 58.3 | 73.7 | 69.7 | 64.4 | 66.6 | 58.7 | 64.5 |
| CoTTA | 60.1 | 62.2 | 60.1 | 73.2 | **62.4** | **72.0** | **74.1** | 66.9 | 68.3 | 59.5 | 75.1 | **72.8** | 67.8 | **72.0** | **66.5** | 67.5 |
| RoTTA | 50.6 | 55.1 | 54.4 | 69.6 | 57.5 | 70.4 | 74.0 | **68.1** | **69.5** | 62.4 | **75.4** | 70.5 | 67.4 | 69.9 | 63.4 | 65.2 |
| SAR | 57.8 | 60.4 | 58.3 | 73.1 | 60.0 | 71.4 | 73.4 | 66.7 | 67.1 | 60.7 | 75.1 | 71.6 | 67.3 | 69.8 | 61.9 | 66.3 |
| ADA | 57.4 | 63.1 | 61.5 | 72.3 | 59.6 | 70.4 | 72.4 | 67.0 | 69.3 | 61.8 | 73.9 | 71.6 | 65.4 | 66.5 | 63.8 | 66.4 |
| ETA | 62.9 | **66.8** | **63.9** | 72.6 | 62.3 | 70.3 | 72.9 | 67.5 | 67.6 | **64.0** | 73.2 | 71.1 | 66.3 | 69.8 | 62.0 | 67.5 |
| ECoTTA | 56.4 | 59.0 | 54.9 | 68.4 | 56.1 | 67.6 | 69.1 | 63.8 | 65.2 | 59.9 | 71.6 | 65.2 | 62.1 | 67.8 | 59.6 | 63.1 |
| Ours | **63.4** | 65.6 | 63.7 | **73.3** | 61.5 | 71.3 | 73.7 | 67.7 | 68.7 | **64.0** | 74.7 | 71.2 | 67.1 | 70.5 | 61.4 | **67.9** |

# L    Results of Semantic Segmentation

## L.1    Results on the Segmentation Task on the CarlaTTA Dataset

In this section, we conduct online continual test-time adaptation experiments using the CARLA simulator [59] across three domain-shift scenarios with varying weather and visual conditions: day-to-night, clean-to-fog, and clean-to-rain. As shown in Table 9a to Table 9c, our method not only adapts well to semantic segmentation tasks, but also consistently outperforms baseline methods across different scenarios.

## L.2    Visualization Results on the Segmentation Task on the Cityscapes Dataset

We further evaluate our method on segmentation tasks using the real-world Cityscapes [60] dataset under corrupted target domains for a more intuitive demonstration of its effectiveness. As shown in Figure 11, while Tent performs well after one epoch of adaptation, its performance deteriorates with prolonged continual adaptation. In contrast, our method maintains stable and accurate segmentation results even after 10 epochs, demonstrating superior robustness in long-term adaptation scenarios.

Table 9: Semantic segmentation results (mIoU/%) on Carla simulation

(a) Results on day2night setting.

| Method | road | sidewalk | building | wall | fence | pole | traffic light | traffic sign | vegetation | terrain | sky | person | vehicle | road line | mIoU |
|---|---|---|---|---|---|---|---|---|---|---|---|---|---|---|---|
| CoTTA | **96.23** | 83.59 | **84.38** | 55.98 | 12.51 | 45.86 | 69.65 | 55.59 | 75.40 | 13.55 | 33.54 | 68.83 | 88.86 | 75.49 | 61.39 |
| Tent | 96.04 | **83.75** | 84.33 | 55.83 | 13.67 | 45.59 | 69.56 | 55.75 | 74.97 | 13.69 | 33.63 | **69.05** | **88.90** | 75.69 | 61.46 |
| Ours | 96.14 | 83.57 | **84.38** | **57.65** | **15.86** | **45.97** | **70.30** | **56.30** | **75.58** | **14.89** | **33.80** | 68.72 | 88.25 | **75.76** | **61.94** |

(b) Results on clear2fog setting.

| Method | road | sidewalk | building | wall | fence | pole | traffic light | traffic sign | vegetation | terrain | sky | person | vehicle | road line | mIoU |
|---|---|---|---|---|---|---|---|---|---|---|---|---|---|---|---|
| CoTTA | 86.13 | **77.33** | **73.05** | **44.49** | 16.66 | **45.82** | 66.01 | 57.00 | 58.96 | 21.67 | 40.28 | 67.49 | **66.35** | **72.03** | 56.60 |
| Tent | 84.71 | 77.00 | 72.16 | 43.42 | 18.59 | 44.37 | 66.11 | 56.50 | 58.57 | 22.24 | **40.98** | 67.23 | 61.07 | 70.71 | 55.97 |
| Ours | **86.94** | 77.13 | 71.65 | 43.59 | **20.20** | 45.34 | **67.00** | **57.31** | **59.27** | **23.57** | 39.57 | **67.93** | 64.06 | 71.62 | **56.89** |

(c) Results on clean2rain setting.

| Method | road | sidewalk | building | wall | fence | pole | traffic light | traffic sign | vegetation | terrain | sky | person | vehicle | road line | mIoU |
|---|---|---|---|---|---|---|---|---|---|---|---|---|---|---|---|
| CoTTA | **95.83** | 87.12 | 90.23 | 72.10 | 22.30 | 54.84 | 81.38 | 65.73 | **81.21** | 21.78 | 70.45 | 75.04 | **90.86** | 80.45 | 70.67 |
| Tent | 95.63 | **87.20** | 90.14 | 72.83 | 26.02 | 54.64 | 81.40 | 65.94 | 80.77 | 22.35 | 69.90 | 75.21 | 90.18 | 80.55 | 70.91 |
| Ours | 95.82 | 87.12 | **90.39** | **73.60** | **27.18** | **55.21** | **81.76** | **66.40** | 80.99 | **23.64** | **71.39** | **75.36** | 90.53 | **80.64** | **71.43** |

(d) Results on dynamic setting.

| Method | road | sidewalk | building | wall | fence | pole | traffic light | traffic sign | vegetation | terrain | sky | person | vehicle | road line | mIoU |
|---|---|---|---|---|---|---|---|---|---|---|---|---|---|---|---|
| CoTTA | 78.99 | 63.81 | 69.22 | 26.79 | 7.96 | 35.46 | 59.16 | 46.20 | 46.79 | 3.37 | 30.20 | 50.32 | **72.08** | 58.47 | 46.34 |
| Tent | 78.56 | 65.95 | 72.82 | 37.88 | 13.01 | 39.79 | 64.26 | 51.60 | 58.24 | 4.17 | 30.20 | 60.28 | 66.62 | 61.44 | 50.35 |
| Ours | **82.21** | **66.06** | **73.41** | **42.96** | **16.96** | **41.95** | **66.80** | **54.20** | **60.44** | **5.76** | **31.27** | **61.35** | 70.12 | **62.38** | **52.56** |

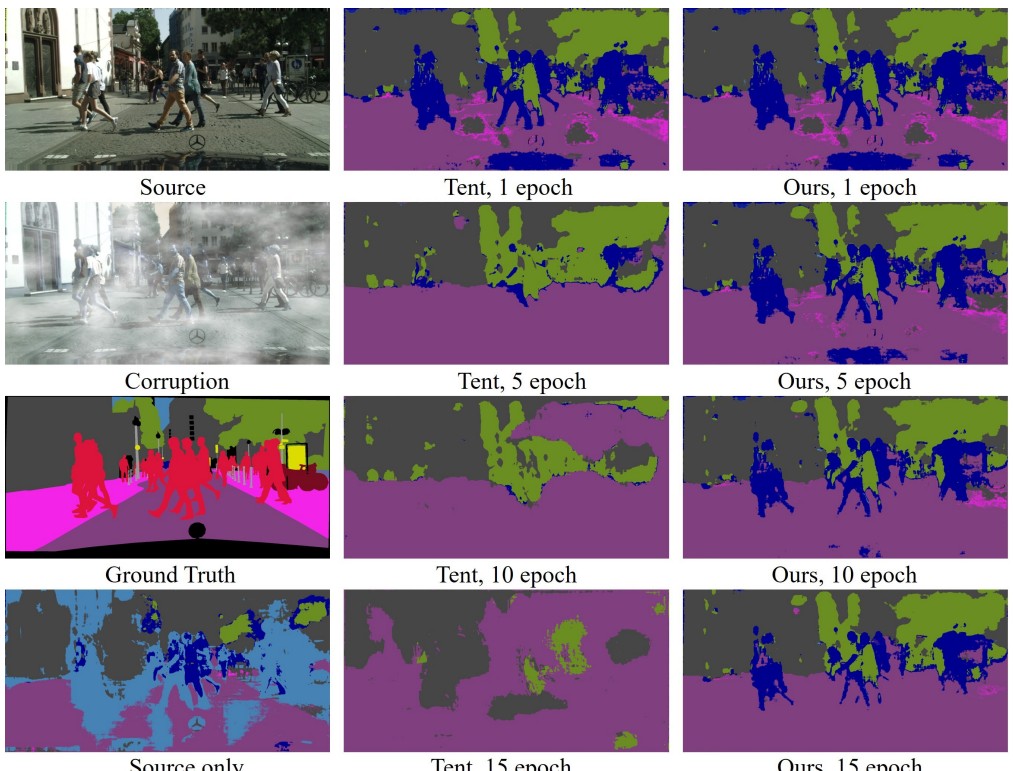

Figure 11: Visualization of segmentation results on the Cityscapes dataset under corruption.

## M  Experimental Setup and Hardware Configuration

We conduct the main experiments of 50 cycles TTA in Section 6.1 on a Linux server equipped with 8 NVIDIA V100 GPUs with 32GB memory each, and an Intel(R) Xeon(R) Platinum 8280 CPU @ 2.70GHz. All other experiments in Section 6.2 are performed on a PC platform equipped with a single Nvidia RTX 3090 GPU with 24GB memory, including the efficiency analysis in Table 7.

