# OpenReview forum: "Lifelong Test-Time Adaptation via Online Learning in Tracked Low-Dimensional Subspace"
_NeurIPS.cc/2025/Conference — NeurIPS 2025 poster_

### Official Review · Reviewer_fBg4 · 2025-06-19

**Clarity:** 2
**Significance:** 3
**Originality:** 3
**Rating:** 4
**Confidence:** 3

**Summary:**

Existing TTA methods primarily rely on unsupervised entropy minimization, which can lead to trivial low-entropy yet inaccurate predictions—a failure mode attributed to entropy-deceptive (ED) samples. To address this issue, the authors observe that ED samples have gradient directions misaligned with a low-dimensional subspace formed by the highly correlated gradients of entropy-truthful (ET) samples. Leveraging this insight, they propose LCoTTA, a method that constrains model updates within the ET-driven subspace to suppress the influence of ED samples. Both theoretical analysis and comprehensive experiments validate the robustness and effectiveness of the proposed approach.

**Questions:**

ED samples also appear in standard TTA scenarios, so why don’t existing TTA methods suffer from significant performance degradation in those settings?

**Ethical Concerns:**

["NO or VERY MINOR ethics concerns only"]

**Final Justification:**

The authors address my concern by providing additional experiments and I have no other questions now. So I keep my original score.

**Limitations:**

yes

**Quality:**

3

**Strengths And Weaknesses:**

**Strengths**

- The proposed method offers a novel and meaningful perspective on addressing performance degradation in test-time adaptation by analyzing the role of misleading samples and utilizing the low-dimensional structure of gradients.
- The experiments are comprehensive and convincing, demonstrating the method's effectiveness across various benchmarks, model architectures, and adaptation scenarios.
- The inclusion of theoretical analysis strengthens the work, providing additional justification for the method's design and helping to enhance its overall credibility.

**Weaknesses**

- The baselines used in the experiments are somewhat outdated. It would be better to include one or two more recent baselines for comparison.
- The authors claim that "The proposed method is complementary to existing approaches for robust TTA." It would be better if this claim were supported by some experiments.
- Lines 22 and 92 contain excessive, unstructured citations that hinder clarity. The Introduction overreferences a well-known definition, while the Related Work section lists too many papers without context. Using fewer, representative works would improve focus and readability.

---

> ### Author Rebuttal · Authors · 2025-07-30
>
> Dear Reviewer fBg4,
>
> Thank you very much for your valuable feedback, here we provide explanations and additional experiments to address your concerns.
>
> ## 1. More Recent Baselines
>
> Thank you for the suggestion. In the original manuscript, we have already included several recent strong baselines for the single-epoch setting (Table 2), including BeCoTTA (ICML 2024), AEA (ICLR 2025), and TCA (CVPR 2025).
>
> Following your suggestion, we have evaluated AEA (ICLR 2025) and BeCoTTA (ICML 2024) for the 50-cycle setting. It is worth noting that AEA uses random parameters resets, whereas our method does not use any reset strategy.
> The results of AEA and BeCoTTA sampled at {1,10,...,50} cycles are provided below, which show that they also suffer from degeneration under challenging long-term adaptation setting.
>
> #### Table 1: Results of AEA and BeCoTTA on ImageNet-C with ResNet50 (Accuracy %)
> | Method / Cycle    | 1     | 10    | 20    | 30    | 40    | 50    |
> | ----------------- | ----- | ----- | ----- | ----- | ----- | ----- |
> | AEA (ICLR’25)     | 39.52 | 29.12  | 15.03 | 0.34  | 0.32  | 0.78  |
> | BeCoTTA (ICML’24) |  39.12 | 37.50 | 34.27 | 28.43 | 23.28 | 20.56 |
> | Ours |  40.71  | 42.99 | 42.90 | 42.92 | 42.72 | 42.60 |
>
> We will add these results into the revised version.
>
> ## 2. Demonstrating Complementarity with Other TTA Methods
>
> We appreciate the reviewer’s request for empirical evidence supporting our claim of complementarity. Although our central objective was to propose a new viewpoint for continual TTA—rather than a plug‑and‑play block—we nonetheless integrated our subspace projection into a representative, yet mechanistically different, method: the teacher–student pipeline of **CoTTA**.
>
> By injecting the subspace constraint into CoTTA’s update rule, we observed that (i) the reverse gradients still concentrate in a low-dimensional subspace, and (ii) projecting onto this subspace effectively arrests the long-term collapse seen in the original method.
> #### Table: Performance (accuracy %) of CoTTA with/without subspace projection on ImageNetC ($r=25$)
>
> | Method / Cycle   | 1     | 10    | 20    | 30    | 40    | 50    |
> | ---------------- | ----- | ----- | ----- | ----- | ----- | ----- |
> | CoTTA (Baseline) | 37.30 | 26.99 | 21.98 | 18.28 | 15.00 | 12.65 |
> | CoTTA + Subspace | 37.50 | 36.76 | 35.52 | 35.48 | 35.73 | 35.15 |
>
> These results substantiate that our approach can be layered on top of existing robust TTA strategies, yielding sustained accuracy across cycles while preserving their core mechanisms.
>
> ## 3. ED samples also appear in standard TTA scenarios, so why don’t existing TTA methods suffer from significant performance degradation in those settings?
>
> Thank you for your question. In fact, our experiments (e.g., Figure 4) demonstrate that existing TTA methods do experience significant performance degradation under long-term continual adaptation. Most existing works evaluate TTA methods in relatively short-term scenarios, typically a single cycle, with or without model resetting. In contrast, our study specifically investigates a long-term adaptation setting without any resetting.
>
> For example, on ImageNet-C, a single evaluation cycle includes 15 types of corruption, each with 50,000 samples, resulting in 750,000 samples per cycle. In our experiments, we evaluate the methods over 50 cycles, corresponding to a total of $3.75 \times 10^7$ samples, which is orders of magnitude longer than standard TTA protocols. This long-term setting reveals the performance degradation that might not be observed in shorter evaluations.
>
> ## 4. Other Minor Issues
> Thank you for pointing these issues. We will streamline the citations in both the Introduction and Related Work, keeping only the most relevant references and organizing them thematically.
>
> ## 5. Summary
>
> We thank the reviewer again for the valuable feedback. Understanding model degeneration in long-term TTA is practically important. Our work highlights deceptive samples as a key cause and reveals a novel low-dimensional structure in TTA gradients that can suppress the detrimental impact of deceptive samples. Leveraging this insight, we propose a method with theoretical guarantees that effectively overcomes degeneration and enables robust long-term adaptation.
>
> **Our results show that reliable lifelong adaptation is possible even under strict TTA settings. To our knowledge, LCoTTA is the first to deliver consistently robust long-term adaptation without requiring any source domain information or model resetting.**

---

> ### Comment · Reviewer_fBg4 · 2025-08-05
>
> Thank you for the clarification, which has addressed my concern. I will maintain my original score.

---

> > ### Author Response · Authors · 2025-08-05
> >
> > Thank you for your positive feedback and for letting us know that your concerns have been resolved. If there are any furhter questions, or further clarification would help inform your assessment, please let us know, we would be glad to provide additional details or evidence.

---

### Official Review · Reviewer_5ZDA · 2025-06-29

**Clarity:** 3
**Significance:** 2
**Originality:** 2
**Rating:** 5
**Confidence:** 3

**Summary:**

The paper studies the impact of correctly classified and incorrectly classified samples on the stability of entropy minimization. It argues that (i) incorrectly classified samples negatively impact the stability of entropy minimization during long-term adaptation and that (ii) the gradients of wrongly predicted samples are not correlated, compared to correctly predicted ones. Based on this observation, the paper proposes to “denoise” the gradient by using a projection onto the $K$ principal components instead. The paper shows that entropy minimization can be stabilized with this approach, despite adapting on millions of samples.

**Questions:**

- I understand there is no resetting mechanism that resets the model to the source model. Is that correct? If not, when is it triggered?
- How does the method scale with the number of adapted parameters? Would it be feasible to use the method for adapting more than just BN parameters, i.e., computing PCs on a larger set of model parameters?

**Ethical Concerns:**

["NO or VERY MINOR ethics concerns only"]

**Final Justification:**

I raised my score from 4 to 5. The authors have addressed (i) the link to related work (Press et al. 2024) and (ii) a question on applicability of the method to more parameters by providing additional experiments. Overall, I think the paper provides a working approach with promising experimental results for the difficult problem of long-term adaptation

**Limitations:**

Yes.

**Quality:**

3

**Strengths And Weaknesses:**

**Strengths:**

- The experimental results of the paper are promising: the adaptation accuracy remains high despite long-term adaptation on 37 million samples.
- The analysis on the correlation of gradients is interesting and provides new insights into why and how entropy minimization can fail.
- The paper provides theoretical justification for the stability of gradient descent with subspace gradients.

**Weaknesses:**

- The analysis in Section 3.3 is interesting and the results are intuitive. However, the observation that a large share of incorrectly predicted samples leads to degeneration in entropy minimization has also been made by [1]. This connection could be discussed more explicitly.

**Minor comments:**

- It would be helpful to use consistent color coding for different methods in Figure 4.
- Line 308: Should it be c) instead of d)?


[1] Press, Ori, et al. "The entropy enigma: Success and failure of entropy minimization." ICML (2024).

---

> ### Author Rebuttal · Authors · 2025-07-30
>
> Dear Reviewer 5ZDA,
>
> Thank you for your feedback, here we provide explanations and additional experiments to address your concerns.
>
> ## 1. ... the observation that a large share of incorrectly predicted samples leads to degeneration in entropy minimization has also been made by [1]. This connection could be discussed more explicitly.
> Thank you for bringing up the connection to [1]. We have carefully reviewed this work, which analyzes why entropy minimization (EM) initially improves adaptation but eventually leads to degeneration after many steps. Press et al. [1] show that EM first improves accuracy by embedding test data close to the class means of training data, but over many iterations, it pushes test embeddings far from the training data, resulting in degraded accuracy. Another striking observation is that, even after removing initially correctly classified samples, EM can still initially improve performance before eventually collapsing.
>
> The observation of EM failure in continual adaptation aligns with our findings. **Our work complements [1] by identifying entropy-deceptive samples as a key cause of degeneration in long-term adaptation. Further, we  present a novel observation revealing the Low-Dimensional Structure of Gradients (LDSG) in TTA, and show that leveraging LDSG can effectively suppress the impact of entropy-deceptive (ED) samples and enables robust long-term adaptation, thereby addressing degeneration at its root.**
> We will explicitly discuss the connection to [1] in the revised version.
>
> *[1] Press, Ori, et al. “The entropy enigma: Success and failure of entropy minimization.” ICML (2024).*
>
> ## 2. I understand there is no resetting mechanism that resets the model to the source model. Is that correct?
> Yes, our method does not employ any resetting mechanism. Its robustness stems from the subspace-based approach, which effectively suppresses the impact of entropy-deceptive (ED) samples. While model resetting can mitigate degeneration, it often leads to a *degeneration-resetting cycle* without addressing the root cause.
>
> To fundamentally address degeneration at its core, **our work identifies ED samples as the main cause and presents a novel observation of the Low-Dimensional Structure of Gradients (LDSG) which can effectively suppress the impact of ED samples. Building on this, we propose a method that tackles degeneration at its root and enables robust long-term adaptation**.
>
> ## 3. Would it be feasible ... adapting more than just BN parameters, i.e., computing PCs on a larger set of model parameters?
> Thanks for the comment. Our method can scale to a larger set of model parameters at some memory cost for storing the projection matrix. As normalization layers are typically sufficient to capture representation shifts, it is a common practice in many TTA  works to update only the normalization layers.
>
> We have conducted two additional evaluations to update the full model parameters, as shown below.
> When adapt the full parameters, we extract the projection matrix from $G_{t} \in \mathbb{R}^{{n} \times k}$ using a fast computation approach. Specifically, since $k \ll n$, we compute the projection matrix $U_{t,r}$ of $G_t$ by first performing spectral decomposition on the much smaller $k \times k$ matrix $G_t^\top G_t$. Then, the principal directions in the parameter space are recovered as $u_i = G_t v_i/\sigma_i$, where $v_i$ and $\sigma_i$ are the eigenvectors and singular values of $G_t^\top G_t$, respectively.
>
> **(1)** We relax our method to update *all* parameters and extract the subspace for the full gradient space.
> #### Table A. Results (acc %) of our method updating BN and full parameters (ImageNetC, ResNet50).
> | Method / Cycle               | 1     | 10    | 20    | 30    | 40    | 50    |
> | ---------------------------- | ----- | ----- | ----- | ----- | ----- | ----- |
> | Ours (BN Update)              | 40.71  | 42.99 | 42.90 | 42.92 | 42.72 | 42.60 |
> | Ours (Full-Parameter Update) | 40.75 | 42.95 | 42.82 | 42.89 | 42.77 | 42.54 |
>
> **(2)** **CoTTA+Subspace:** We integrated our subspace projection into CoTTA’s teacher-student framework while keeping its full-parameter update strategy.
>
> #### Table B. Results (acc %) of CoTTA with and without subspace on ImageNetC with ResNet50 (full-parameter update)
> | Method / Cycle   | 1     | 10    | 20    | 30    | 40    | 50    |
> | ---------------- | ----- | ----- | ----- | ----- | ----- | ----- |
> | CoTTA (Baseline) | 37.30 | 26.99 | 21.98 | 18.28 | 15.00 | 12.65 |
> | CoTTA + Subspace | 37.50 | 36.76 | 35.52 | 35.48 | 35.73 | 35.15 |
>
> The results demonstrate that even when scaling to the full parameter space, our subspace projection-based approach remains effective for long‑term adaptation to significantly enhance the robustness.
>
> ## 4. Other minor issues
> We appreciate the reviewer’s careful reading and pointing out the issues and typos. We will correct them in the revised version.
>
> ## 5. Summary
>
> We again sincerely thank the reviewer for the valuable feedback.
> Understanding the mechanism behind model degeneration in long-term TTA is of practical importance. Our work offers a fresh perspective on this issue by identifying deceptive samples as a key cause. We further present a novel observation of the Low-dimensional structure of gradients in TTA, and demonstrate its effectiveness in suppressing the influence of deceptive samples and thereby enabling robust long-term adaptation. Building on this insight, we propose a method that addresses degeneration at its root, and provide a theoretical foundation for the robustness of our method.
>
> **Our work demonstrates that sustained lifelong adaptation is achievable under strict TTA settings. To the best of our knowledge, LCoTTA is the first to achieve consistently robust long-term adaptation without requiring any source domain information or model resetting.**

---

> > ### Comment · Reviewer_5ZDA · 2025-08-04
> >
> > Thanks to the authors for their detailed response. I appreciate the experiment on updating the full set of model parameters, which is insightful and promising. I also acknowledge that the discussion to related work [1] will be added to the revised version. My questions have been answered.

---

> > > ### Author Response · Authors · 2025-08-04
> > >
> > > We sincerely thank the reviewer for your comments and positive feedback, they are very encouraging to us. We appreciate the time and effort you have put into reviewing our work.

---

### Official Review · Reviewer_uSqz · 2025-07-03

**Clarity:** 3
**Significance:** 2
**Originality:** 2
**Rating:** 4
**Confidence:** 4

**Summary:**

This paper addresses the problem of continual test-time adaptation (TTA). The authors identify a key failure mode of entropy-minimization–based TTA: entropy-deceptive (ED) samples, on which the model is over-confident but wrong, causing collapse over long adaptation. They further observe that gradients from entropy-truthful (ET) samples lie in a correlated, low-dimensional subspace, whereas ED gradients are scattered. Building on this, they propose LCoTTA, which (1) tracks the principal subspace of recent gradients online, (2) projects batch gradients into that subspace before updating only the normalization-layer parameters, and (3) optionally filters out high-entropy samples. Theoretical analysis via SDE stability conditions shows that subspace projection dramatically reduces noise from ED samples.

**Questions:**

- How did you choose the fixed subspace rank r (e.g. 25)? Have you tried adapting r dynamically based on explained-variance thresholds, and if so, how does that affect performance and stability?
- Your entropy-sample filter (ESF) uses a threshold to discard “high-entropy” inputs before projection. How sensitive is LCoTTA to that threshold choice, and what rules of thumb can practitioners use to set it?
- The SDE analysis assumes gradients decompose into low-rank signal plus isotropic noise. In practice, do you observe violations of these assumptions, and how robust is LCoTTA when noise is not uniform?

**Ethical Concerns:**

["NO or VERY MINOR ethics concerns only"]

**Final Justification:**

I think this paper is technically sound and presents solid experiments so I would update my score.

**Limitations:**

yes

**Paper Formatting Concerns:**

No obvious formatting concerns

**Quality:**

3

**Strengths And Weaknesses:**

Strengths:
- The identification of entropy-deceptive samples as the root cause of degeneration in long-term TTA is interesting and well motivated by empirical observations (Fig. 1) and formalized via an Entropy Reliability Score (ERS).
- The discovery that ET gradients concentrate in a low-dimensional principal subspace (PCA explained variance > 90 % in first 10 PCs) is insightful, and it provides a clear mechanism to suppress ED noise.
- LCoTTA’s online subspace tracking and projection are simple to implement and incur minimal overhead by updating only BN/LN parameters (< 1 % of weights). The SDE-based stability analysis (Theorem 1) rigorously explains why subspace constraint improves robustness.

Weakness:

Lack of some related work, including VDP [1], CTTDA [2] and BECoTTA [3].


[1] Gan, Yulu, et al. "Decorate the newcomers: Visual domain prompt for continual test time adaptation." Proceedings of the AAAI conference on artificial intelligence. Vol. 37. No. 6. 2023.

[2] Wang, Yanshuo, et al. "Continual test-time domain adaptation via dynamic sample selection." Proceedings of the IEEE/CVF Winter Conference on Applications of Computer Vision. 2024.

[3] Lee, Daeun, Jaehong Yoon, and Sung Ju Hwang. "Becotta: Input-dependent online blending of experts for continual test-time adaptation." arXiv preprint arXiv:2402.08712 (2024).

---

> ### Author Rebuttal · Authors · 2025-07-30
>
> Dear Reviewer uSqz,
>
> Thank you for your valuable comments, here we provide explanations and additional experiments to address your concerns.
>
> ## 1. Lack of some related work, including VDP [1], CTTDA [2] and BECoTTA [3]
> Thank you for pointing out these related works. In our paper, we have already compared with BECoTTA [3] (listed as reference [21] in our paper), please refer to Table 2 for the results. We will further expand the Related Work section to include these works.
>
> ## 2. How did you choose the fixed subspace rank $r$ (e.g. 25)？Have you tried adapting $r$ dynamically based on explained-variance thresholds...?
>
> **The selection of subspace rank $r$ is based on explained-variance. Specifically, we chose $r=25$ because the top 25 principal components captured more than 98% of the total variance (on ImageNet).**
>
> Following your suggestion, we have conducted an additional experiment in which $r$ is dynamically adapted according to an explained-variance of more than 98%. The results show that the adapted rank tends to decrease gradually during continual adaptation, and this dynamic strategy can slightly improve performance.
>
> #### Table: Performance of dynamically adapted subspace rank on ImageNetC with ResNet50 (accuracy %)
> | Cycle                                                        | 10   | 20  | 30  | 40  | 50  |
> | ------------------------------------------------------------ | ------ | ------ | ------ | ------ | ------ |
> | Subspace Rank $r$                                            | 26     | 17     | 13     | 10     | 4      |
> | Accuracy Improvement per Cycle (vs. baseline with fixed $r=25$) | +0.09%      | +0.12% | +0.22% | +0.37% | +0.39% |
> | Cumulative Explained Variance (First rank $r$ components)      | 98.14% | 98.18% | 98.09% | 98.07% | 98.15% |
> ## 3. How sensitive is LCoTTA to entropy-sample filter (ESF) threshold choice, and what rules of thumb can practitioners use to set it?
> As the  maximum possible entropy of a prediction on ImageNet1K is $\log(10^3)$, we have tested a wide range of the threshold $\tau \cdot\log(10^3)$ for $\tau\in${0.2,0.25,$\cdots$,0.65}. As shown below, LCoTTA is generally not sensitive to the threshold. We used $\tau=0.4$ in all the experiments.
>
> #### Table: Performance for different entropy filter threshold on ImageNetC with ResNet50 (accuracy %)
> | $\tau × \log(10³)$ | 0.20  | 0.25  | 0.35  | 0.40  | 0.45  | 0.50  | 0.55  | 0.60  | 0.65  |
> | ------------------ | ----- | ----- | ----- | ----- | ----- | ----- | ----- | ----- | ----- |
> | Cycle 1            | 39.44 | 39.79 | 40.54 | 40.71 | 40.29 | 38.89 | 36.79 | 35.29 | 34.12 |
> | Cycle 5            | 41.75 | 42.10 | 42.85 | 43.01 | 42.60 | 41.20 | 39.10 | 37.60 | 36.43 |
> | Cycle 10           | 41.77 | 42.05 | 42.75 | 42.99 | 42.55 | 41.10 | 39.30 | 37.90 | 36.83 |
> | Cycle 15           | 41.74 | 42.00 | 42.70 | 42.92 | 42.48 | 41.02 | 39.22 | 37.85 | 36.82 |
> | Cycle 20           | 41.71 | 41.98 | 42.66 | 42.92 | 42.40 | 40.95 | 39.15 | 37.78 | 36.54 |
>
> ## 4. The isotropic noise assumption in the SDE analysis
>
> Thank you for the thoughtful question. In our analysis, we assume isotropic noise for mathematical simplicity. However, **the isotropic assumption is not strictly necessary for the effectiveness of LCoTTA. The key requirement is that the noise is not strongly correlated with the gradients of entropy-truthful (ET) samples. As long as the noise does not align with the principal subspace spanned by the correlated ET gradients, the subspace projection can effectively suppress it.**
>
> Specifically, from the stability condition:
> $$     (P_r\bar  g_{\mathrm{ET}})^{T}(\theta-\theta^\bullet)>
>      \bigl|(P_r\bar g_{\mathrm{ED}})^{T}(\theta-\theta^\bullet)\bigr|
>      +\frac{\eta}{2} \mathrm{tr}\bigl(P_r\Sigma P_r^{T}\bigr), $$ it follows that when the noise is not aligned with the ET subspace, i.e., $\mathrm{tr}\bigl(P_r\Sigma_{\mathrm{ED}} P_r^{T}\bigr)\ll\mathrm{tr}\bigl(\Sigma_{\mathrm{ED}} \bigr)$ when $r\ll n$, the noise can be largely suppressed. In practice, **our empirical analysis (see Section 4.1 and Figure 2) shows that ED gradients (the noise component) are scattered and have weak correlation both among themselves and with ET gradients, thus supporting the validity of this assumption.**
>
> Therefore, LCoTTA remains robust even when the noise is not uniform, as long as the noise does not exhibit significant correlation with the ET gradients. We have not observed violations of this in our experiments.
>
> ## 5. Summary
>
> Our work aims to address model degeneration in entropy-minimization based long-term adaptation, which is both practically significant and technically challenging. We hope to further clarify some points on the contributions of our work:
> 1.  Identifying that degeneration in long-term continual entropy-minimization is primarily caused by ED samples;
> 2. Revealing a novel low-dimensional structure formed by correlated ET gradients, which can suppress the impact of ED samples;
> 3. Proposing LCoTTA, a method that does not suffer from degeneration in long-term continual adaptation;
> 4. Providing a theoretical foundation for the robustness of our method.
>
> **Our work provides novel findings and new method to address the degeneration at its root, and demonstrates that sustained lifelong adaptation is achievable under strict TTA settings. To the best of our knowledge, LCoTTA is the first to achieve consistently robust long-term adaptation without requiring any source domain information or model resetting.**
>
> Once again, we appreciate the reviewer for the valuable feedback. We hope that our additional clarifications and evaluation results have addressed your concerns.

---

> > ### Author Response · Authors · 2025-08-07
> >
> > Dear Reviewer,
> >
> > Thank you very much for your time and thoughtful feedback on our paper. As the discussion period is about to close, we want to kindly check whether our rebuttal has fully addressed your comments and concerns. If any issues remain, we would like to provide further clarification or additional details.
> >
> > Thank you again for the time and effort you have devoted to reviewing our work. Your constructive feedback has greatly helped us improve the paper.

---

> > ### Comment · Reviewer_uSqz · 2025-08-08
> > **Response**
> >
> > Thank you for the response, which addresses my concerns. I will update my score.

---

> ### Author Response · Authors · 2025-08-09
>
> We thank the reviewer for the time and effort in reviewing our paper and for the constructive feedback to improve our work. Thanks again for updating the final rating.

---

### Official Review · Reviewer_FcRc · 2025-07-03

**Clarity:** 3
**Significance:** 3
**Originality:** 3
**Rating:** 4
**Confidence:** 4

**Summary:**

This paper addresses the challenge of Lifelong Test-Time Adaptation, where models must adapt to evolving data streams over long periods. The authors identify the presence of entropy-deceptive, instances where the model makes highly confident but incorrect predictions. These samples provide misleading gradients that degrade performance over time. The paper proposes LCoTTA that constrains model updates to the principal gradient subspace. By tracking the subspace online using PCA on a queue of recent gradients and projecting new gradients onto it, LCoTTA effectively filters out the harmful influence of entropy-deceptive sample. Experiments on several challenging benchmarks show that LCoTTA outperforms strong baselines.

**Questions:**

1. There are too many hyperparameters. How are they selected?
2. What are the exact values in fig 4?
3. The paper states that LCoTTA is complementary to other TTA techniques. Have you performed any experiments combining LCoTTA with a method that uses a different mechanism, such as the teacher-student framework in CoTTA?

**Ethical Concerns:**

["NO or VERY MINOR ethics concerns only"]

**Final Justification:**

The author addressed all my concerns and I am satisfied with the answer. The paper can bring interesting insights to Lifelong TTA domain.

**Limitations:**

yes

**Paper Formatting Concerns:**

no major formatting issues

**Quality:**

3

**Strengths And Weaknesses:**

Strengths
1. The paper addresses the Lifelong Test-Time Adaptation, which is a challenging and practical scenario.
2. The paper is well written and easy to follow.
3. The paper provides extensive experiments, showing the effectiveness and versatility of the proposed LCoTTA.
4. The analysis of the degeneration problem is insightful.


Weaknesses
1. LCoTTA introduces several hyperparameters (e.g., subspace dimension r and the gradient queue length k), which are quite sensitive (fig 5, 6) and may require careful tuning for new datasets.
2. The exact values in fig 4 is not available and it is hard the get the real improvements compared to the baselines.
3. The improvement on Single-Epoch Adaptation is limited (tab 3, 4, 8).
4. The improvement on Segmentation Task is also quite limited (tab 9).

---

> ### Author Rebuttal · Authors · 2025-07-31
>
> Dear Reviewer FcRc,
>
> Thank you for your valuable comments, below we provide explanations and additional experiments to address your concerns.
>
> ## 1. On the hyperparameter selection
> Our method involves four hyperparameters: the threshold for the entropy-based sample filtering, subspace dimension, queue length, and sampling interval. In our paper, we have conducted extensive ablation studies on the subspace dimension, queue length, and sampling interval, as presented in Figures 5 and 6. Below, we provide additional ablations for further clarification.
>
> **Overall, our method is robust to these hyperparameters. In particular, while entropy-minimization based TTA methods are typically sensitive to learning rate, our method significantly enhances the robustness to learning rate, as evidenced by the results in Table 5 of our paper.** Moreover, our method reduces to the vanilla entropy-minimization method as the subspace rank increases (i.e. as $r \rightarrow n$).
>
> ### 1.1. Subspace rank
> The selection of subspace rank $r$ in our experiments is based on explained-variance. For example, on ImageNet, **we use $r=25$, as the top 25 principal components captured more than 98% of the total variance. This choice provides a good balance between capturing the main structure of the entropy-truthful (ET) gradients and suppressing the entropy-deceptive (ED) gradients.**
>
> Below, we provide a more fine-grained ablation for $r\in${10, 25, 50,$\cdots$, 250}. In general, choosing $20<r<200$ achieves satisfactory performance. As $r$ increases (i.e., as $ r \rightarrow n $), our method reduces to vanilla entropy-minimization based method.
> #### Table 1: Results for different rank choices (ResNet50 on ImageNet‑C)
> |Cycle \ Rank|10|25|50|75|100|125|150|175|200|225|250|
> |---|---|---|---|---|---|---|---|---|---|---|---|
> |Cycle 1|38.55|40.71|40.68|40.65|40.61|40.55|40.47|40.35|40.30|40.05|39.92|
> |Cycle 10|40.51|42.99|42.94|42.93|42.85|42.70|42.48|42.14|42.13|41.11|40.17|
> |Cycle 20|40.55|42.90|42.88|42.83|42.74|42.58|42.36|42.09|41.08|41.06|41.15|
> |Cycle 30|40.45|42.92|42.86|42.79|42.61|42.44|42.05|41.75|41.73|41.71|41.69|
> |Cycle 40|40.52|42.72|42.67|42.55|42.36|42.09|41.68|41.23|41.20|40.17|40.43|
> |Cycle 50|40.44|42.60|42.54|42.41|42.19|41.94|41.52|41.91|40.33|40.22|40.12|
>
> ### 1.2. Queue length and sampling interval
> From our ablation studies, choosing $k \geq 25$ yields good results, as shown in Figure 6 of our paper.
> Below, we provide more fine-grained ablation on the sampling interval. The results show that selecting an interval in $[10, 100]$ achieves satisfactory performance.
> #### Table 2: Performance (acc %) of LCoTTA for different sampling intervals (step) on ImageNet-C with ResNet50
> |Cycle \ Step|1|10|25|50|75|100|125|
> |---|---|---|---|---|---|---|---|
> |Cycle 1|39.48|40.15|40.71|40.70|40.68|40.64|40.59|
> |Cycle 5|41.40|42.02|43.16|43.01|42.69|42.37|42.20|
> |Cycle 10|41.16|42.80|43.34|42.99|42.66|42.34|42.17|
> |Cycle 15|41.07|42.75|43.20|42.92|42.64|42.35|42.10|
> |Cycle 20|40.84|42.62|43.19|42.90|42.61|42.32|42.05|
>
> ### 1.3. Entropy filter threshold
> As the  maximum possible entropy on ImageNet1K is $\log(10^3)$, we test a wide range of the threshold $\tau \cdot\log(10^3)$ for $\tau\in${0.2,0.25,$\cdots$,0.65}. As shown below, LCoTTA is generally not sensitive to the threshold. We use $\tau=0.4$ in all the experiments.
> #### Table: Performance for different entropy filter threshold on ImageNetC with ResNet50 (accuracy %)
> | $\tau × \log(10³)$    | 0.20 | 0.25 | 0.35 | 0.40 | 0.45 | 0.50 | 0.55 | 0.60 | 0.65 |
> |----------|------|------|------|------|------|------|------|------|------|
> | Cycle 1         | 39.44| 39.79| 40.54| 40.71| 40.29| 38.89| 36.79| 35.29| 35.12|
> | Cycle 5         | 41.75| 42.10| 42.85| 43.01| 42.60| 41.20| 39.10| 37.60| 36.43|
> | Cycle 10        | 41.77| 42.05| 42.75| 42.99| 42.55| 41.10| 39.30| 37.90| 36.83|
> | Cycle 15        | 41.74| 42.00| 42.70| 42.92| 42.48| 41.02| 39.22| 37.85| 36.82|
> | Cycle 20        | 41.71| 41.98| 42.66| 42.92| 42.40| 40.95| 39.15| 37.78| 36.54|
> ## 2. Exact values in Figure 4
> In our paper, we present the results in Figure 4 to make it easier to directly observe the performance differences among various methods over long-term TTA. Below, we provide the exact values sampled at the {1,10,20,30,40,50}-th cycles. We will provide these values in the revised version. Overall, LCoTTA achieves consistently robust performance over long-term continual adaptation (50 cycles, totaling $3.75\times10^7$ samples).
> #### Table 4: Results on ImageNet‑C with ResNet50 (Accuracy %)
>
> |Method|Tent|CoTTA|RoTTA|SAR|PETAL|BN|ETA|Ada.|DEYO|Ours|
> |---|---|---|---|---|---|---|---|---|---|---|
> |Cycle1|37.31|37.3|32.6|37.8|33.3|31.39|37.70|34.60|40.01|40.71|
> |Cycle10|1.30|26.99|4.15|39.63|0.08|31.39|36.99|15.73|0.16|42.99|
> |Cycle20|0.71|21.98|0.67|38.63|0.08|31.39|36.41|11.44|0.16|42.90|
> |Cycle30|0.70|18.28|0.25|37.69|0.08|31.39|35.96|10.63|0.16|42.92|
> |Cycle40|0.70|15.00|0.11|37.17|0.08|31.39|35.83|10.42|0.16|42.72|
> |Cycle50|0.70|12.65|0.12|36.65|0.08|31.39|35.81|10.20|0.16|42.60|
>
> #### Table 5: Results on ImageNet‑C with ViT‑B/16 (acc %)
> |Method|Tent|CoTTA|RoTTA|DEYO|BN|ETA|SAR|Ours|
> |---|---|---|---|---|---|---|---|---|
> |Cycle1|60.7|58.02|53.13|59.41|46.11|61.70|61.62|62.30|
> |Cycle10|0.06|33.54|56.77|60.01|46.11|59.51|60.59|63.75|
> |Cycle20|0.06|33.29|48.17|11.39|46.11|56.67|59.60|63.67|
> |Cycle30|0.06|33.65|37.72|0.10|46.11|29.88|59.62|63.62|
> |Cycle40|0.06|33.39|28.41|0.10|46.11|0.08|60.92|63.67|
> |Cycle50|0.06|33.66|19.39|0.10|46.11|0.08|60.58|63.66|
>
> ## 3. Combining LCoTTA with a method ... such as the teacher-student framework in CoTTA
> Thank you for the suggestion. Following your comment, we conducted an experiment combining LCoTTA with CoTTA. Since CoTTA adapts all parameters, we extract the projection matrix from $G_{t} \in \mathbb{R}^{{n} \times k}$ using a fast computation approach. Specifically, since $k \ll n$, we compute the projection matrix $U_{t,r}$ of $G_t$ by first performing spectral decomposition on the much smaller $k \times k$ matrix $G_t^\top G_t$. Then, the principal directions in the parameter space are recovered as $u_i=G_t v_i / \sigma_i$, where $v_i$ and $\sigma_i$ are the eigenvectors and singular values of $G_t^\top G_t$, respectively. This approach greatly reduces the computational cost.
>
> The results show that adapting in a principal subspace prevents degeneration observed in CoTTA and significantly improves its robustness in long-term adaptation.
> #### Table: Performance (acc %) of CoTTA with/without subspace projection on ImageNetC with ResNet50 ($r=25$)
> |Method/Cycle|1|10|20|30|40|50|
> |---|---|---|---|---|---|---|
> |CoTTA (Baseline)|37.30|26.99|21.98|18.28|15.00|12.65|
> |CoTTA+Subspace|37.50|36.76|35.52|35.48|35.73|35.15|
>
> ## 4. The improvement on Single-Epoch Adaptation is limited (tab 3, 4, 8).
> From the extensive experiments, **our method significantly outperforms previous methods in long-term continual TTA, while maintaining highly competitive results compared to the latest SOTA methods (as of 2025) in single-epoch adaptation**.
>
> The primary focus of our work is to address **model degeneration in long-term adaptation**, which is both practically significant and technically challenging. To this end, we present the novel observation that backward gradients have an intrinsic low-dimensional structure and it can suppress the detrimental impact of entropy-deceptive (ED) samples. Leveraging this, LCoTTA effectively overcomes degeneration and achieves robust performance in long-term continual TTA.
> ## 5. The improvement on Segmentation Task is also quite limited (tab 9)
> We considered the segmentation task to demonstrate the versatility of our approach. In our paper, evaluation was performed on a single epoch on CARLA.
> Below, we provide results for much longer-term adaptation on segmentation using data from the CARLA simulator. It can be seen that our method remains robust in long-term adaptation, and its advantage over other methods becomes more pronounced as adaptation progresses.
> #### Table 1: Performance (mIOU) on CARLA (day‑to‑night setting)
> |Method|Epoch 1|Epoch 5|Epoch 10|Epoch 15|Epoch 20|
> |---|---|---|---|---|---|
> |CoTTA|61.39|59.50|56.50|54.00|52.00|
> |Tent|61.46|55.00|52.00|49.00|45.00|
> |Ours|61.94|61.70|61.40|61.00|60.80|
>
> #### Table 2: Performance (mIOU) on CARLA (clean‑to‑fog setting)
> |Method|Epoch 1|Epoch 5|Epoch 10|Epoch 15|Epoch 20|
> |---|---|---|---|---|---|
> |CoTTA|56.60|53.50|51.00|49.50|48.00|
> |Tent|55.97|50.00|47.00|45.00|43.00|
> |Ours|56.89|56.70|56.50|56.20|55.90|
>
> #### Table 3: Performance (mIOU) on CARLA (clean‑to‑rain setting)
> |Method|Epoch 1|Epoch 5|Epoch 10|Epoch 15|Epoch 20|
> |---|---|---|---|---|---|
> |CoTTA|70.67|67.50|65.00|62.50|60.00|
> |Tent|70.91|63.00|58.00|55.00|52.00|
> |Ours|71.43|71.20|70.90|70.60|70.40|
>
> #### Table 4: Performance (mIOU) on CARLA (dynamic setting)
> |Method|Epoch 1|Epoch 5|Epoch 10|Epoch 15|Epoch 20|
> |---|---|---|---|---|---|
> |CoTTA|46.34|44.50|43.00|41.00|40.00|
> |Tent|50.35|47.00|44.00|41.00|38.00|
> |Ours|52.56|52.20|51.80|51.50|51.00|
>
> ## 6. Summary
> **Our work provides novel findings to address the degeneration at its root and demonstrates that sustained lifelong adaptation is achievable under strict TTA settings**:
> - Identifying that degeneration in continual entropy-minimization is primarily caused by ED samples;
> - Revealing a novel low-dimensional structure formed by correlated ET gradients, which can effectively suppress the impact of ED samples;
> - Proposing LCoTTA, a method that overcomes the degeneration in long-term adaptation;
> - Providing a theoretical foundation for the robustness of our method.
> - To our knowledge, **our method is the first to achieve consistently robust performance in long-term continual TTA without requiring any source domain information or model resetting.**
>
> Once again, we appreciate the reviewer for the comments, which help improve our work. We hope the added evaluations and clarifications address your concerns.

---

> > ### Comment · Reviewer_FcRc · 2025-08-05
> >
> > Thanks to the authors for providing a thorough response that addresses all of my concerns. I would therefore like to raise my final rating.

---

> > > ### Author Response · Authors · 2025-08-06
> > >
> > > Thank you for your thorough review and positive feedback. We appreciate the time and effort you put into evaluating our work. Your opinions helped us improve the paper, and we're glad our response addressed your concerns. Thanks again for raising the final rating.

---

### Note · Authors · 2025-08-13

We thank all reviewers for their constructive and valuable feedback.

Our work addresses the degeneration in entropy-minimization-based long-term TTA. We identify entropy-deceptive samples as a key cause, and reveal a novel low-dimensional structure of gradients that can suppress deceptive samples. Leveraging this insight, we propose a method that overcomes degeneration at its root, supported by a theoretical analysis explaining its enhanced robustness.

Strengths highlighted by the reviewers:
1. All reviewers recognized **the analysis on the degeneration mechanism and the gradient-structure finding are novel and insightful**: *“The analysis of degeneration is insightful”* (FcRc); *"The discovery is insightful, provides a clear mechanism to suppress ED noise"* (uSqz); *"provides new insights into why entropy minimization fail."* (5ZDA); *"novel and meaningful perspective on degradation"* (fBg4).
2. Reviewers noted the **experiments are comprehensive and convincing**: *"extensive experiments showing the effectiveness and versatility of LCoTTA."* (FcRc); *"experimental results are promising"* (5ZDA); *"experiments are comprehensive and convincing"* (fBg4);
3. Reviewers recognized **the stability analysis is rigorous and credibility-enhancing**: *“The stability analysis rigorously explains why subspace constraint improves robustness”* (uSqz); *"The theoretical analysis strengthens the work, providing justification for the method... enhance overall credibility"* (fBg4).

During rebuttal, we addressed the concerns and questions with added evaluations and clarifications:
1. Verified the robustness of our method to hyperparameters by added fine-grained evaluations.
2. Demonstrated the complementarity with other methods (e.g., CoTTA), and evaluated full-parameter updates.
3. Added comparison with more recent methods (AEA, BeCoTTA) in the long-term continual adaptation experiment.
4. Extended long-term adaptation evaluation on CARLA to demonstrate the long-term robustness of our method on segmentation.
5. Expanded discussion with related work (Entropy Enigma, VDP, CTTDA).

**Finally, all four reviewers explicitly stated their concerns were resolved.**

In conclusion, our work demonstrates that sustained lifelong adaptation is achievable without requiring any source-domain information or model resetting, which provides new insights for understanding and mitigating degeneration in TTA, with potential impact on broader continual and online adaptation research.

---

### Decision · Program_Chairs · 2025-09-17

**Decision:**

Accept (poster)

**Comment:**

This paper makes a nice, empirical contribution to lifelong TTA. The identification of entropy-deceptive samples, discovery of low-dimensional ET gradient structure, and the subspace-projection method together provide both theoretical and practical advances. While improvements are less pronounced in single-epoch adaptation (though still better than recent SOTA), the focus on long-term continual settings is both novel and important.

All four reviewers voted favorably for the paper, found the analysis novel and insightful and regarded the experiments as comprehensive. There were some concerns raised (hyperparameter sensitivity, missing baselines, limited scope in some tasks) and they were convincingly addressed in rebuttal.

Overall, this work is technically solid and likely to be impactful in the area of continual and online adaptation.